# Relativistic Bohmian trajectories of photons via weak measurements

Joshua Foo [1 ✉], Estelle Asmodelle [1], Austin P. Lund [1,2] & Timothy C. Ralph[1 ✉]

Bohmian mechanics is a nonlocal hidden-variable interpretation of quantum theory which predicts that particles follow deterministic trajectories in spacetime. Historically, the study of Bohmian trajectories has mainly been restricted to nonrelativistic regimes due to the widely held belief that the theory is incompatible with special relativity. Here, we present an approach for constructing the relativistic Bohmian-type velocity field of single particles. The advantage of our proposal is that it is operational in nature, grounded in weak measurements of the particle's momentum and energy. We apply our weak measurement formalism to obtain the relativistic spacetime trajectories of photons in a Michelson–Sagnac inter-ferometer. The trajectories satisfy quantum-mechanical continuity and the relativistic velocity addition rule. We propose a modified Alcubierre metric which could give rise to these trajectories within the paradigm of general relativity.

[1] Centre for Quantum Computation & Communication Technology, School of Mathematics & Physics, The University of Queensland, St. Lucia, QLD 4072, Australia. [2] Dahlem Center for Complex Quantum Systems, Freie Universität Berlin, 14195 Berlin, Germany. ✉email: joshua.foo@uqconnect.edu.au; ralph@physics.uq.edu.au

Ever since the conception and early development of quantum mechanics, there have been debates over how to consistently interpret the mathematical objects it defines. Many of these debates, such as those concerning the physical meaning of the wavefunction[1–3], continue to this day. The predominant view among present-day physicists is the Copenhagen interpretation, which treats the square of the wavefunction,

$$\rho_S(x,t) = |\psi(x,t)|^2, \qquad (1)$$

as the probability density of finding a particle at the spacetime point $(x, t)$, a postulate known as the Born rule. Thus, the Copenhagen interpretation is, by its very definition, an intrinsically probabilistic formulation of quantum theory. This indeterminacy is also manifest in the Heisenberg uncertainty principle, which forbids observations that would permit simultaneous knowledge of the precise position and momentum of a particle.

In contrast with standard interpretations of quantum theory, Bohmian mechanics is a deterministic, nonlocal theory first proposed by de Broglie[4] and then formalised by Bohm[5,6]. As described in Bell's famous paper, it is a theory of the nonlocal hidden-variable type[7,8]. Specifically, the wavefunction (itself evolving according to the Schrödinger equation) determines the evolution of a nonlocal guiding potential which in turn, governs the dynamics of the particles. Notably, the Bohmian interpretation recovers many of the standard results of nonrelativistic quantum mechanics such as the Born rule, where instead of describing the probability distribution of a particle's location, $\rho_S(x, t)$ is now interpreted as the density of particle trajectories given some initial distribution.

Bohmian mechanics has been applied to a diverse number of nonrelativistic settings. Early works by Hirschfelder et al.[9,10], Philippidis et al.[11], and Dewdney et. al.[12] focused on the scattering of massive particles off different kinds of potential barriers. Bohm, Dewdney et al. and Durr et al. extended the theory to include spin-1/2 particles and provided a description of EPR correlations[13–15]; Leavens applied it to the measurement of arrival times[16], while others have utilised Bohmian mechanics in the context of quantum chaos and interference[17–19].

Interest in Bohmian mechanics was reinvigorated following the development of a weak measurement model by Wiseman[20], from which an operational definition for the velocity and hence the spacetime trajectory of nonrelativistic particles can be determined (by operational, we mean a precise, step-by-step set of instructions using only fundamental concepts such as measurement and time-evolution). Weak measurements were first proposed by Aharonov, Albert, and Vaidman[21] as a method of performing arbitrarily precise measurements of quantum-mechanical observables with minimal system disturbance. A weak measurement of an observable $\hat{a}$ is one which only weakly perturbs the system of interest, but concomitantly carries a large amount of measurement uncertainty. Performing repeated weak measurements on an ensemble scales this uncertainty as $1/\sqrt{N}$ (where $N$ is the number of measurements) allowing one to estimate the average value $\langle \hat{a} \rangle$ with arbitrarily high precision.

A 'weak value' extends this notion of weak measurement by introducing a subsequent strong measurement and performing post-selection on the ensemble based on this strong measurement. Formally, the weak value of $\hat{a}$, denoted $_{\langle \phi |}\langle \hat{a}_w \rangle_{|\psi\rangle}$, is the mean value of $\hat{a}$ obtained from many weak measurements on an ensemble of particles each prepared in the state $|\psi\rangle$, postselecting only those particles where a later strong measurement reveals the system to be in the state $|\phi\rangle$. This results in the following definition[1]:

$$_{\langle \phi |}\langle \hat{a}_w \rangle_{|\psi(t)\rangle} = \mathrm{Re}\,\frac{\langle \phi | \hat{a} | \psi(t)\rangle}{\langle \phi | \psi(t)\rangle}. \qquad (2)$$

From this starting point, Wiseman was able to connect the nonrelativistic Bohmian velocity field with the notion of weak values[20].

Using Wiseman's weak measurement framework, two landmark experiments by Kocsis et al.[22] and Mahler et al.[23] were able to infer the Bohmian trajectories of nonrelativistic particles by constructing a velocity field in terms of a momentum weak value:

$$V(x,t) = \frac{\langle x | \langle \hat{p}_w \rangle_{|\psi(t)\rangle}}{m}, \qquad (3)$$

where $|x\rangle$ is a position eigenstate, $m$ is the effective mass of the particle and $|\psi(t)\rangle$ is the initial state of the wavefunction. Equation (3) can itself be deduced from the nonrelativistic definition of velocity:

$$V(x,t) = \frac{p_x}{m}. \qquad (4)$$

The above mentioned experiments highlight the explanatory power of Wiseman's formulation of nonrelativistic Bohmian mechanics; it is a manifestly operational framework which utilises a measurement formalism to construct the Bohmian velocity field, from which the resulting particle trajectories can be inferred. Although the existence of deterministic particle trajectories in Bohmian mechanics contrasts the more conventional probabilistic interpretation of the wavefunction, the underlying predictions are consistent between both perspectives.

Despite its successes in the nonrelativistic domain, a formidable challenge continues to face the proponents of Bohmian mechanics, namely its apparent conflict with special relativity. Many prior studies have demonstrated the difficulties and interpretive issues in constructing a physically meaningful theory for relativistic scalar particles, particularly when utilising the Klein–Gordon equation as a starting point[6,24]. Indeed it is well-known that the time-component of the conserved four-current density in the Klein–Gordon theory is not positive-definite, raising concerns about its ability to be understood as a probability density[6,25–29], and whether particle trajectories in such scenarios are manifestly causal[30–32]. In his treatment of electromagnetic scattering, Bohm asserted that a particle description of photons is fundamentally inconsistent, insisting that a field description is necessary[25]. Flack and Hiley have raised concerns that a relativistic treatment of photon trajectories is likely unphysical due to the existence of reference frames in which the photon's velocity is zero[33]. Other works have used the relativistic Dirac equation as the basis for constructing a Bohmian theory for spin-1/2 particles[13,31,34], however, these studies are not without their own issues. For example, Nikolic's formulation requires the postulation of additional hidden-variables, whose physical interpretation is not clear.

The interpretive issues common to these studies arise because they consistently take as a starting point various theories of or modifications to relativistic, single-particle quantum mechanics, which already possess the aforementioned pathologies. The missing link between these studies and a consistent interpretation of relativistic Bohmian mechanics is the notion of operationalism, that is the ability to ground phenomena in the measurement of physical observables. In this sense, a consistent, operationally-based theory of Bohmian particle trajectories constructed from observed velocity fields in relativistic regimes has not been developed.

In this article, we propose a method of constructing the Bohmian velocity field of relativistic particles, specifically photons possessing a relativistic energy dispersion. Our proposed velocity equation is defined operationally through weak measurements of the particle momentum and energy, and may thus be understood as a reformulation of Wiseman's nonrelativistic framework. Indeed, we show that a naïve application of Wiseman's weak measurement definition in the relativistic limit fails due to tacit nonrelativistic assumptions in his construction, most notably a

privileged timeslice on which measurements of the particle position are made. Importantly, our expression for the velocity field reduces to that predicted by Wiseman's nonrelativistic theory in the paraxial (low-energy) limit. From our weak measurement definition, we make use of the inherited relativistic properties of the Klein–Gordon wavefunction in order to evaluate the Bohmian trajectories of photons in a Michelson–Sagnac-type interferometer. This point of contact with the Klein–Gordon theory captures the essential Lorentz relativistic properties of photons with the understanding that this is a simplification of a full electrodynamical theory. Crucially, by grounding these trajectories in ensemble measurements, we are able to make a unique and internally consistent interpretation of the trajectories which arise. The derived velocity field is Lorentz covariant under boosts, equivalently satisfying the relativistic velocity addition rule. Our analysis focuses on the 'optical' limit where the distribution of wavevectors is concentrated well away from zero. This is consistent with our measurement-based framework; if the optical approximation is not satisfied, one enters the subcycle regime (where measurement timescales are shorter than a single wavelength) where particle creation and annihilation processes become evident. That is, the domain of applicability of our theory is one in which a single-particle description is valid (hence the optical approximation implies a single-particle description). We finally draw a connection between the relativistic wavefunction and the Alcubierre metric[35], proposing this as a relativistic generalisation to the quantum potential[5] which is usually understood as guiding the particle in the nonrelativistic limit.

Throughout this article, we utilise natural units, $\hbar = c = 1$ and the metric signature $(-, +, +)$ in (2+1)-dimensional Minkowski spacetime.

## Results

**Physical setup**. The physical geometry of our system, represented as a Michelson–Sagnac interferometer, is depicted in Fig. 1 on a background of flat spacetime in (2+1)-dimensions. A controllable beamsplitter diverts the path of the single photon to the top or bottom branch of the interferometer. Of course within a conventional interpretation of quantum mechanics, the photon traverses both arms of the interferometer in superposition, whereas in Bohmian mechanics the photon is guided by the pilot wave along one of the paths.

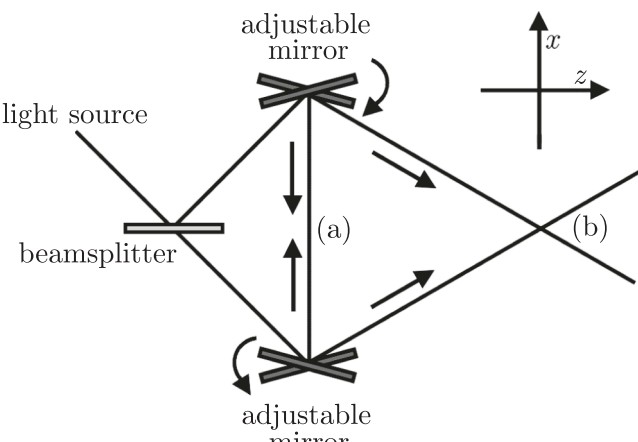

**Fig. 1 Schematic diagram illustrating the geometry of the setup.** The adjustable mirrors control the path of the photon in the transverse direction, which can be tuned to produce a head-on collision (**a**) or a grazing collision (**b**).

The adjustable mirrors reflect the photon towards the centre of the apparatus. In scenario (a), the wavevector in the $z$-direction is approximately zero, $k \equiv k_x \gg k_z$, which we refer to as a head-on collision. Scenario (b) depicts the general case in which the photon possesses a non-zero wavevector component in the $z$-direction, which we refer to as the relativistic grazing scenario. When $k_z \gg k$, the $x$-component of the photon velocities are slow; this nonrelativistic regime is known as the paraxial limit. In the literature, only the paraxial limit has been studied in analogous setups.

**Relativistic Bohmian velocity field from weak measurements**. Unlike prior studies[6,27,30–32], our starting point for the derivation of a relativistic Bohmian theory is not relativistic single-particle quantum mechanics (e.g. the Klein–Gordon equation for scalar particles). Instead, we base our construction on an operational foundation, beginning with the notion of weak measurements, reformulating Wiseman's nonrelativistic framework. Consider first the relativistic extension to the classical velocity field of Eq. (4):

$$V(x, t) = \frac{\mathrm{d}x}{\mathrm{d}t} = \frac{\mathrm{d}x}{\mathrm{d}\tau} \cdot \frac{\mathrm{d}\tau}{\mathrm{d}t} = \frac{p_x}{E} \quad (5)$$

(upon restoring factors of $c$, Eq. (5) is a dimensionless velocity). In our analysis, we focus on the particle dynamics in the $x$-direction, in accordance with the setup shown in Fig. 1. Meanwhile, variables in the $z$-direction are treated as constants of the problem. Equation (5) is a relativistic quantity; the coordinate time, $t$, and position, $x$, form a relativistic spacetime two-vector $\mathbf{x} = (t, x)$, as do the total energy of the photon and the $x$ component of the momentum; $\mathbf{p} = (E, p_x)$. These components transform Lorentz covariantly under boosts, by construction. Crucially, the velocity field is a coordinate velocity; we need not make reference to the proper time $\tau$ defined along a given trajectory.

Equipped with the toolkit of weak values and in view of this relativistic definition of the velocity, we propose the following definition for the relativistic velocity field of single photons:

$$V(x, t) = \frac{\langle x | (\hat{k}_w)_{|\psi(t)\rangle}}{\langle x | (\hat{H}_w)_{|\psi(t)\rangle}} = \frac{\mathrm{Re}\langle x | \hat{k} | \psi(t)\rangle}{\mathrm{Re}\langle x | \hat{H} | \psi(t)\rangle}. \quad (6)$$

Equation (6) is the main result of our paper, namely a Bohmian-type velocity field for relativistic particles constructed from a purely operational vantage point. In Eq. (6), $\hat{k} \equiv \hat{k}_x$ is the momentum operator (modulo a multiplicative factor of $\hbar$ set to unity) while $\hat{H}$ is the Hamiltonian. The weak values of these operators are defined with respect to an initial preparation for the ensemble of particles in the state $|\psi\rangle$ and projected onto the single-particle position eigenstate $|x\rangle$ (i.e. postselection at the spacetime point $(x, t)$),

$$|x\rangle = \int \mathrm{d}k\, e^{-ikx} |k\rangle. \quad (7)$$

It must be emphasised that the relativistic velocity field in Eq. (6) is a coordinate velocity constructed from an in-principle measurement in a particular reference frame. When the weak measurements are used to construct trajectories in a given reference frame, one can consistently apply the Lorentz transformation to obtain the velocity field in a different reference frame. Conversely, if one transforms into a different frame and applies this operational definition, the resulting velocities are related by a standard Lorentz transformation.

Let us return to our proposed velocity equation. We are interested in plotting sample particle trajectories in the relativistic regime, arising from the measurement-based velocity field of Eq. (6). We consider the particle in the state

$$|\psi\rangle = \int \mathrm{d}k\, f(k) |k\rangle \quad (8)$$

where $f(k) \equiv f(k_x)$ determines the initial distribution of the particle momenta, and $\hat{a}_k^\dagger|0\rangle = |k\rangle \equiv |k_x\rangle$ is a single-particle momentum eigenstate. In order to evaluate Eq. (6), we utilise the differential relationships generated from the scalar Klein–Gordon field momentum and Hamiltonian, and project onto $|x\rangle$ to find that Eq. (6) reduces to

$$V(x,t) = \frac{2\,\mathrm{Im}\,\psi^\star(x,t)\partial^x\psi(x,t)}{2\,\mathrm{Im}\,\psi^\star(x,t)\partial^t\psi(x,t)}. \tag{9}$$

where $\psi(x,t) = \langle x|\psi(t)\rangle$ is the time-dependent position-space wavefunction. We have utilised the scalar Klein–Gordon theory to capture—in the simplest way—the relativistic properties of the field, however an extension to spin-1/2 or spin-1 particles can be achieved by using the relevant elements (such as a vector-valued wavefunction) from the spin-1/2 Dirac or spin-1 Klein–Gordon theories[36].

In Eq. (9), we readily identify the numerator and denominator of the velocity equation with the conserved probability current and density obtained from the single-particle Klein–Gordon equation:

$$V(x,t) = \frac{j_K(x,t)}{\rho_K(x,t)} \tag{10}$$

where

$$j_K^\mu(x,t) = 2\,\mathrm{Im}\,\psi^\star(x,t)\partial^\mu\psi(x,t) \tag{11}$$

are the components of the Klein–Gordon conserved current vector. These components are relativistic four(two)-vector quantities, and are thus Lorentz covariant by construction[37]. While we have begun with a measurement-based interpretation of the particle trajectories, our connection between this operational model and the relativistic components of the Klein–Gordon conserved current means that the velocity field inherits the desired relativistic properties of the theory. These components also satisfy a continuity equation $\partial_\mu j_K^\mu(x,t) = 0$, or explicitly in terms of components in a chosen coordinate system

$$\frac{\partial \rho_K(x,t)}{\partial t} = -\frac{\partial j_K(x,t)}{\partial x}. \tag{12}$$

The Lorentz covariance of our velocity equation allows us to consider the particle trajectories from a boosted reference frame. Consider an observer moving with velocity $v$ relative to the laboratory frame. According to the relativistic velocity addition rule, the velocity equation in the Lorentz boosted frame takes the form

$$V'(x,t) = \frac{V(x,t) - v}{1 - vV(x,t)}. \tag{13}$$

Recalling that $V(x,t) = j_K(x,t)/\rho_K(x,t)$, we obtain

$$V'(x,t) = \frac{\gamma(j_K(x,t) - v\rho_K(x,t))}{\gamma(\rho_K(x,t) - vj_K(x,t))} = \frac{j_K'(x,t)}{\rho_K'(x,t)} \tag{14}$$

where $\gamma = 1/\sqrt{1-v^2}$ is the Lorentz factor. This illustrates how the velocity equation possesses an identical form in both the lab and boosted reference frames, signifying the Lorentz covariance of our theory. The resulting velocity field, Eq. (10) is the relativistic version of that obtained from the nonrelativistic Schrödinger equation,

$$V_S(x,t) = \frac{j_S(x,t)}{\rho_S(x,t)}, \tag{15}$$

where $j_K(x,t) = (1/k_z)\,\mathrm{Im}\,\psi_S^\star(x,t)\partial^x\psi_S(x,t)$ is the nonrelativistic Schrödinger current and $\rho_S(x,t)$ is the probability density of Eq. (1). One can arrive at Eq. (15) using Wiseman's nonrelativistic formalism.

**Bohmian photon trajectories in the head-on limit**. We can now calculate Bohmian-style particle trajectories in the relativistic, head-on limit. These trajectories can be obtained by integrating the velocity equation, Eq. (5), yielding a parametric function of the spacetime coordinates $(x,t)$. We firstly consider a head-on collision in which the dispersion relation can be approximated by $E(k) = \sqrt{k^2 + k_z^2} \simeq |k|$, that is $k_z \ll k$. We assume the initial distribution of the frequencies to be a superposition of left- and right-moving Gaussian wavepackets

$$f_R(k) = \mathcal{N}_R \exp\left[-\frac{(k - k_{0R})^2}{4\sigma_R^2}\right], \tag{16}$$

$$f_L(k) = \mathcal{N}_L \exp\left[-\frac{(k + k_{0L})^2}{4\sigma_L^2}\right], \tag{17}$$

where $\mathcal{N}_L$ and $\mathcal{N}_R$ are normalisation constants, $k_{0L}$, $k_{0R} > 0$ are the centre frequencies of the left- and right-moving parts of the wave, and $\sigma_L, \sigma_R > 0$ are the variances. The wavefunction is thus given by

$$\psi_K(x,t) = \sqrt{\alpha}\int dk f_R(k)e^{-i|k|t + ikx} + \sqrt{1-\alpha}\int dk f_L(k)e^{-i|k|t + ikx}, \tag{18}$$

where $0 \le \alpha \le 1$. For simplicity, let us assume that the left- and right-moving wavepackets are centred at $-k_0$ and $+k_0$ respectively with equal variance $\sigma_L = \sigma_R \equiv \sigma$. We focus our analysis on a regime known as the optical limit, for which the left- and right-moving wavepackets only have support on negative and positive values of $k$ respectively; that is, the frequency of the light beam is much larger than its spread, $k_0 \gg \sigma$. Finally, we assume that our photon position detectors, modelled as projections onto the $x$-eigenstate $|x\rangle$, are highly resolved in position space.

Using Eq. (18), we find the following expressions for the Klein–Gordon probability current,

$$j_K(x,t) = \alpha\sqrt{\frac{2}{\pi}}\sigma\exp[-2(t-x)^2\sigma^2] \\ - (1-\alpha)\sqrt{\frac{2}{\pi}}\sigma\exp[-2(t+x)^2\sigma^2] \\ + \sqrt{\alpha(1-\alpha)}\sqrt{\frac{2}{\pi}}\mathcal{S}_0\sigma\exp[-2(t^2+x^2)\sigma^2] \tag{19}$$

and density,

$$\rho_K(x,t) = \alpha\sqrt{\frac{2}{\pi}}\sigma\exp[-2(t-x)^2\sigma^2] \\ + (1-\alpha)\sqrt{\frac{2}{\pi}}\sigma\exp[-2(t+x)^2\sigma^2] \\ + 2\sqrt{\alpha(1-\alpha)}\sqrt{\frac{2}{\pi}}\mathcal{T}_0\,\sigma\exp[-2(t^2+x^2)\sigma^2] \tag{20}$$

where both equations have been normalised by $2k_0$, the standard relativistic normalisation factor of the Klein–Gordon theory, and we have defined the following functions of $(x,t)$:

$$\mathcal{S}_0 = 4\frac{\sigma^2 t}{k_0}\sin(2k_0 x), \tag{21}$$

$$\mathcal{T}_0 = \cos(2k_0 x) - 2\frac{\sigma^2 x}{k_0}\sin(2k_0 x). \tag{22}$$

Equations (19) and (20) are inserted into Eq. (10) to obtain the velocity field of the particle. In the limits $\alpha = 0$ or $\alpha = 1$, one

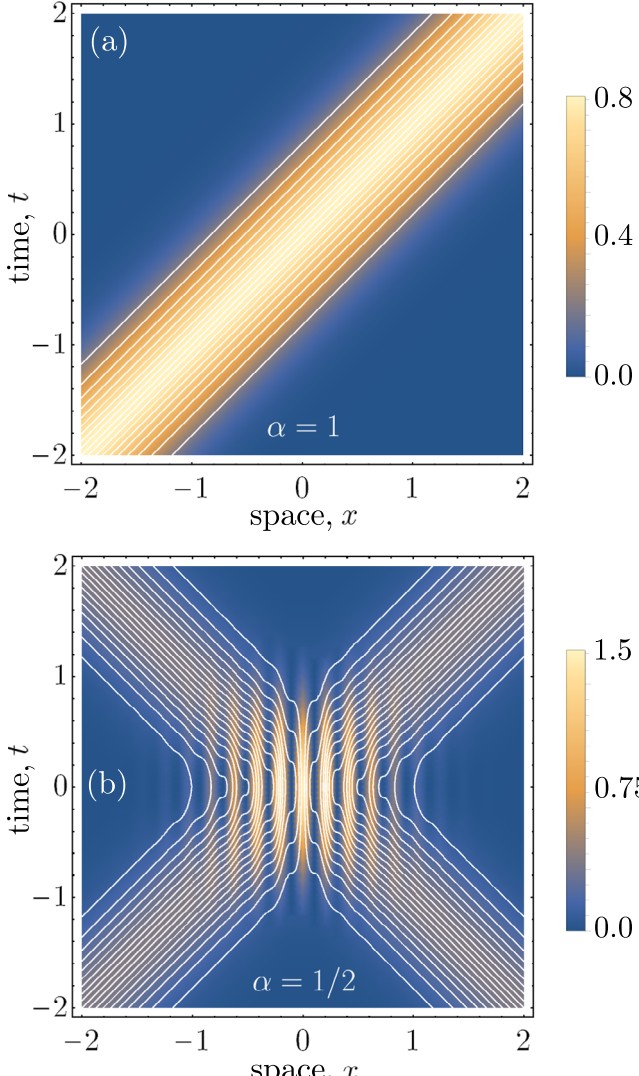

**Fig. 2 Plots of the Bohmian trajectories with $k_0/\sigma = 15$.** In the background of both plots is the quantum-mechanical prediction for the probability density of the particles, $\rho_K(x, t)$. In (**a**), the trajectories are completely right-moving and maintain a Gaussian distribution, according to the initial conditions that we have chosen to match the wavefunction density. In (**b**), we have considered the superposition case, wherein the photon follows a path that matches the interference pattern predicted by $\rho_K(x, t)$.

obtains an entirely left- or right-moving wavepacket respectively, that is, $V(x, t) = \mp 1$. Meanwhile for $0 < \alpha < 1$, the non-zero cross terms emerge, introducing interference fringes in the probability density.

Before analysing the particle trajectories, it must be noted that Eq. (20) is not positive definite, and thus generally cannot be interpreted as a probability density. Regions of negative density emerge when the last term of $\mathcal{T}_0$ becomes non-negligible. However since our analysis is focused on the optical regime where $k_0 \gg \sigma$, such that the magnitude of this term is small compared with the first, $\rho_K(x, t)$ remains positive definite and can be interpreted as a true probability density. Indeed, such a regime is necessitated by the assumptions of our model, which is explicitly a single-particle theory. When $k_0 \sim \sigma$, one leaves the domain of applicability of the single-particle limit, as we elaborate upon in the discussion. Finally, if one naively takes the density to be the modulus square of the wavefuncion, $|\psi(x, t)|^2$, one obtains Eq. (20) without the final term in $\mathcal{T}_0$.

In Fig. 2, we have plotted the Bohmian trajectories for the particles in the head-on collision scenario for different weightings of the superposition parameter $\alpha$. The trajectories for different initial conditions (spaced in proportion to the wavepacket density) are shown in white, superimposed on top of the Klein–Gordon density. In Fig. 2a, the particles are moving solely in the right-moving direction. As expected for relativistic massless particles, the trajectories maintain a Gaussian profile without dispersion. Importantly, the density of trajectories corresponds exactly with the Klein–Gordon probability density, and this matching condition holds for all time. This is an essential consistency requirement of Bohmian mechanics, and the continuity equation plays a crucial role in ensuring this condition.

Likewise in the equal superposition case, the density of trajectories matches exactly with the interference pattern predicted by $\rho_K(x, t)$. In regions of constructive interference the trajectories bunch up, while the opposite occurs for regions of destructive interference. Notably, some of the trajectories become superluminal in regions of destructive interference. As we discuss in section "The photon metric", this is consistent with a relativistic generalisation of the Bohmian quantum potential.

In Fig. 3, we have plotted the Bohmian trajectories in the boosted coordinates of a moving observer. The regions of interference are now tilted with respect to these coordinates, since the observer is moving past the apparatus at a relativistic velocity. Due to the covariance of the continuity equation under boosts, the density of trajectories is conserved, matching the quantum-mechanical prediction.

**Bohmian photon trajectories in the relativistic grazing regime.** We can also consider the general case with energy dispersion given by $E(k) = \sqrt{k^2 + k_z^2}$. If $k_z$ is constant and small but non-zero, this represents a grazing collision regime shown in Fig. 1b. Since there is no simple expression in terms of elementary functions for the wavefunction integrals, we utilise a numerical analysis for this case.

Figure 4 displays the Bohmian trajectories in the grazing relativistic regime. A notable feature is the dispersion of the trajectories as they propagate in time. This is due to the inclusion of the nonzero effective mass term $k_z$ in the particle energy.

**Bohmian photon trajectories in the paraxial limit.** To obtain the nonrelativistic limit of our velocity equation, we consider the regime $k_z \gg k$, so that the total energy takes the form $E(k) \simeq k_z + k^2/2k_z$. As in the relativistic grazing regime, $k_z$ can be interpreted as a mass-like term. The velocity equation under this approximation becomes

$$V(x, t) = \frac{1}{2k_z} \frac{2 \, \mathrm{Im} \, \psi_S^\star(x, t) \partial^x \psi_S(x, t)}{|\psi_S(x, t)|^2} \tag{23}$$

where the wavefunction is explicitly

$$\psi_S(x, t) = \sqrt{\alpha} e^{-ik_z t} \int dk \, e^{ikx - \frac{ik^2 t}{2k_z}} f_R(k) + \sqrt{1-\alpha} e^{-ik_z t} \int dk \, e^{ikx - \frac{ik^2 t}{2k_z}} f_L(k). \tag{24}$$

The numerator and denominator of Eq. (23) reduce to the nonrelativistic versions of the conserved current and density,

$$V(x, t) = \frac{j_S(x, t)}{\rho_S(x, t)}, \tag{25}$$

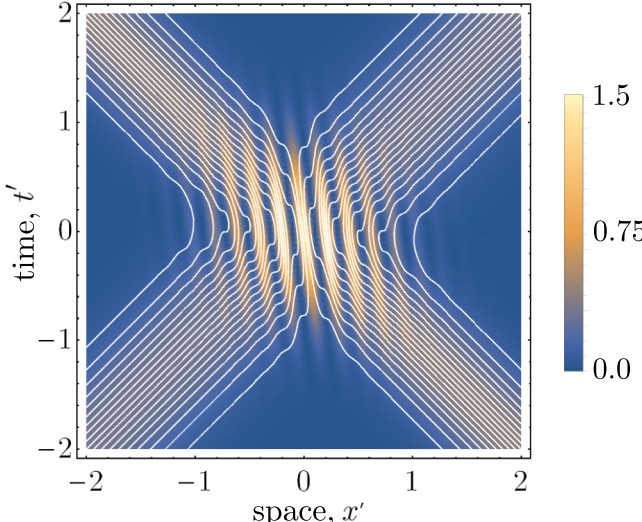

**Fig. 3 Plot of the Bohmian trajectories with respect to the boosted coordinates of an observer who is moving at $v = 0.125$ with respect to the lab frame.** We have used the parameters $k_0/\sigma = 15$ and $\alpha = 1/2$ as in Fig. 2.

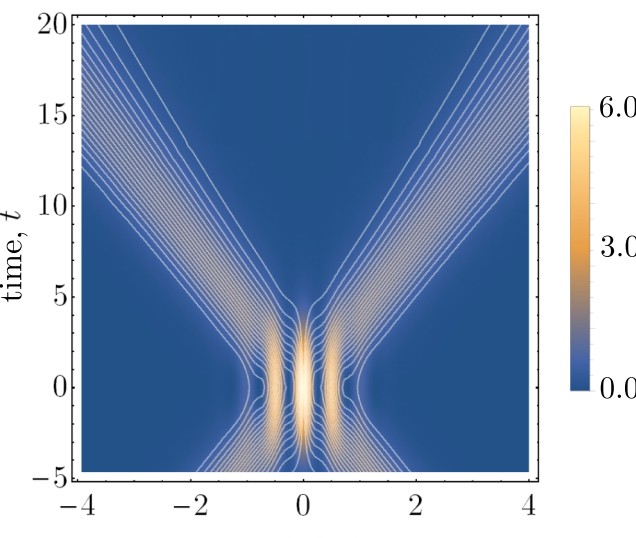

**Fig. 4 Bohmian trajectories for a generic relativistic dispersion with $k_0/\sigma = 6$, $k_z/\sigma = 24$, and $\alpha = 1/2$.** The effect of $k_z$ is to introduce some dispersion in the trajectory density as the particles propagate in time.

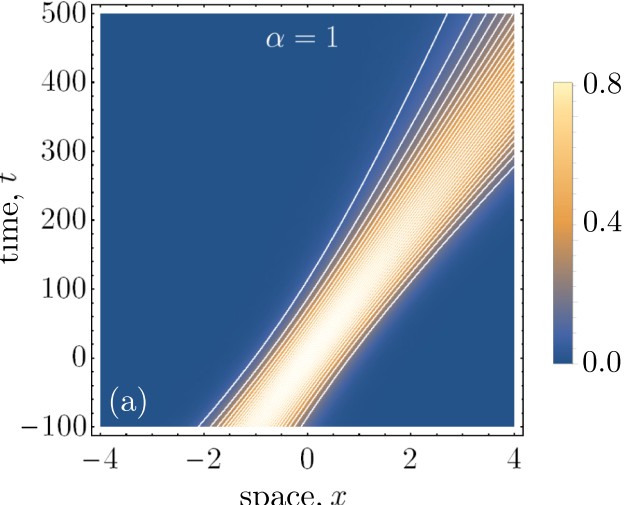

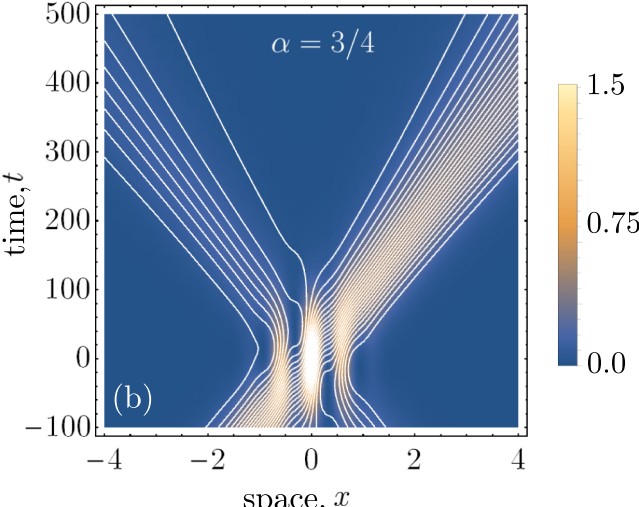

**Fig. 5 Plots of the Bohmian trajectories with $k_0/\sigma = 5$, $k_z/\sigma = 500$.** As in the relativistic grazing limit, the particles exhibit dispersion as they propagate in time. Notably, this occurs on a much larger timescale as compared with the relativistic cases.

producing a velocity field that transforms in the appropriate nonrelativistic way under Galilean boosts. Notably, Eq. (23) yields the same result as that derived by Wiseman, who used the following definition for the velocity in the nonrelativistic, paraxial limit[20]:

$$V(x,t) = \lim_{\delta t \to 0} \frac{1}{\delta t}[x - {}_{\langle x|} \langle \hat{x}_w \rangle_{|\psi(t)\rangle}]. \tag{26}$$

Wiseman's approach in obtaining Eq. (26) was to perform a weak measurement of the particle position at time $t$, followed by a strong measurement of the position a short time $\delta t$ later. The velocity is obtained by calculating the rate of change of position over this small time interval, $\delta t$. When extending this equation to relativistic particles, it ultimately fails due to the introduction of

this Lorentz non-covariant parameter in the evolution between the weak and strong measurements. If the relativistic limit $E(k) \simeq |k|$ is naively applied to Eq. (26), it can be shown that the continuity equation is not satisfied in general.

Nevertheless, Eqs. (23) and (26) are valid expressions for the Bohmian velocity of nonrelativistic particles. In Fig. 5, we have plotted these trajectories in the paraxial limit. Like the relativistic grazing scenario, the particles exhibit dispersion as they propagate in time. The slow-moving particle velocities cause the regions of interference to stretch out in time, comparing the timescales of Figs. 2 and 5.

**The photon metric.** A key concept introduced in Bohm's theory is the quantum potential, $Q(x,t)$, which appears as an additional term in the so-called 'quantum Hamilton–Jacobi' equation obtained from the Schrödinger equation[6]. In the paraxial limit for our single-photon system, the quantum potential takes the form

$$Q(x,t) = -\frac{\hbar^2}{2k_z}\frac{\nabla^2 R(x,t)}{R(x,t)}, \tag{27}$$

where $R^2(x, t) = |\psi(x, t)|^2$. Using Eq. (27), one can obtain the quantum analogue of a classical force, using the standard definition $F_Q(x, t) = -\nabla Q(x, t)$, and this ultimately governs the dynamics of the particle. Evidently, the wavefunction plays a crucial role in determining $Q(x, t)$. When extending Bohmian mechanics to relativistic regimes as we have done here, the 'quantum force' obtained from the quantum potential may now be interpreted as being generated by a quantum spacetime metric. In this way, $\psi(x, t)$ still plays a role in governing the effective dynamics of the particle, but now by defining the shortest path through the spacetime in the paradigm of general relativity.

The problem is that the dynamics generated by such a metric must allow for trajectories that have both spacelike and timelike tangents. A candidate for the kinds of head-on particle trajectories observed in Figs. 2 and 3 is the Alcubierre metric, given by

$$ds^2 = -(1 - v_s^2)dt^2 - 2v_s\,dx\,dt + dx^2 \tag{28}$$

where we have suppressed the $dz$ component for simplicity. Equation (28) was proposed by Alcubierre in ref. [35] as a so-called warp drive solution to Einstein's field equations. In such a spacetime, particles can achieve superluminal velocities through a local expansion of spacetime behind them and a simultaneous contraction of the spacetime in front of them. As is the case with other non-hyperbolic metrics like wormholes[38,39], exotic matter (i.e. negative energy density[40]) is required to produce such effects. Nevertheless, Eq. (28) is a consistent solution within the framework of general relativity, and by defining

$$v_s = \left(\left|\frac{j_K(x, t)}{\rho_K(x, t)}\right| - 1\right)\text{sgn}\left(\frac{j_K(x, t)}{\rho_K(x, t)}\right) \tag{29}$$

where sgn($x$) represents the function giving $+1$ on positive values and $-1$ on non-positive values of $x$, the speed of light according to a faraway observer in an asymptotically flat spacetime region is given by

$$c = v_s + \text{sgn}\left(\frac{j_K(x, t)}{\rho_K(x, t)}\right) = \frac{j_K(x, t)}{\rho_K(x, t)}. \tag{30}$$

This is exactly the prediction of our proposed relativistic Bohmian theory, wherein the coordinate velocity of the particle can exceed the speed of light (i.e. in regions of destructive interference), but also equal zero; that is, when the photon stops and changes directions. Of course, such behaviour is permissible within the paradigm of general relativity, which only requires that light always travels at the speed of light locally. That such a metric exists in the presence of our unusual trajectories could be useful for later work when considering couplings (e.g. conformal coupling) in a field theory, should such an extension to our current single-particle theory be considered.

## DISCUSSION
In this article, we have extended the theory of single-particle Bohmian mechanics to include particles with relativistic energies, via a reformulation of Wiseman's weak measurement formalism and shown that such a description is consistent with Lorentz covariance. This represents an important step in testing the limits of the Bohmian interpretation, which to date has been believed to break down in relativistic regimes.

As emphasised throughout this work, we have employed an optical approximation in our analysis, wherein the frequency of the incoming wavepackets is much larger than their variance, $k_0 \gg \sigma$. In this regime, the Klein–Gordon density remains strictly positive, allowing for its interpretation as a probability density. However, it is well-known that the scalar Klein–Gordon density

$\rho_K(x, t)$ can become negative in certain regimes. As noted previously, the matching condition between the density of Bohmian trajectories and the probability density of the guiding wave holds for all time, and this is true even in regions of negative density. In these regions, the tangent vector to the trajectory becomes negative, yielding particle trajectories that travel backwards in time. The negative density of these backward-in-time trajectories matches the regions of 'negativity' in the wavefunction density, demonstrating the mathematical consistency of our theory in these regimes.

There are two physically distinct regimes in which negativity in the density may arise. The first occurs when one leaves the optics approximation, which is the domain of applicability for our single-particle theory. When $k_0 \sim \sigma$, the low-frequency tail of the wavepacket $f(k)$ impinges significantly into the 'negative-frequency' region of momentum space, which is typically associated with the particle creation operator in quantum field theory[41]. One can also enter this regime by performing measurements at timescales shorter than a single wavelength of light, inducing stimulated particle production from the vacuum. Obtaining a consistent physical interpretation of trajectories in the subcycle regime likely requires an extension of our theory to a full field-theoretic description of the Bohmian mechanics that can handle states with non-conserved particle number. In such a theory, one would require a more complete measurement model which is valid outside the optics limit and incorporates particle production effects. However, the mathematical consistency of our single-particle relativistic theory may give some guidance in such an endeavour.

The second regime in which backwards-in-time trajectories emerge occurs when the particle trajectories are viewed from the reference frame of a rapidly boosted observer, Fig. 6. Unlike the prior regime, such trajectories are physically consistent within our construction, since the redshifted frequency and bandwidth of the photon wavepackets still satisfy the optics approximation in the new reference frame (see "Methods" for a further discussion). When the Klein–Gordon density in the boosted frame equals zero, Eq. (20), the particle velocity according to an observer in this frame,

$$V'(x, t) = \frac{V(x, t) - v}{1 - vV(x, t)}, \tag{31}$$

diverges. Indeed, this property is inevitable for superluminal particle velocities satisfying the velocity addition rule. The divergence of the velocity occurs for pairs of spacetime points; as shown in Fig. 6, the particle 'enters' the region of negative density at infinite velocity, travels backwards in time for some distance, before 'exiting' this region at another point with infinite velocity. It is crucial to distinguish the emergence of negatively directed trajectories due to the non-satisfaction of the optics approximation, as compared with those arising due to boosts. The latter case, as alluded to in Eq. (31), is a consequence of the frame-dependent property of the derived velocity field; in general relativity, one can always obtain locally (and in this case, e.g. Fig. 6, globally) a frame where the velocity field is forward directing (a proof of this is shown in section "Obtaining the reference frame in which $\rho(x, t)$ is positive definite"). The trajectories shown in Fig. 6 are thus entirely consistent with our construction of $V(x, t)$ as a coordinate velocity obtained in a particular reference frame. Operationally, this is also consistent with the well-known property of weak values which can yield 'anomalous' results such as negative energies[42–44].

As mentioned, Bohmian trajectories have been inferred experimentally for photons which are transversely slow (i.e. the paraxial limit)[22,23]. In those experiments, the velocity field of the

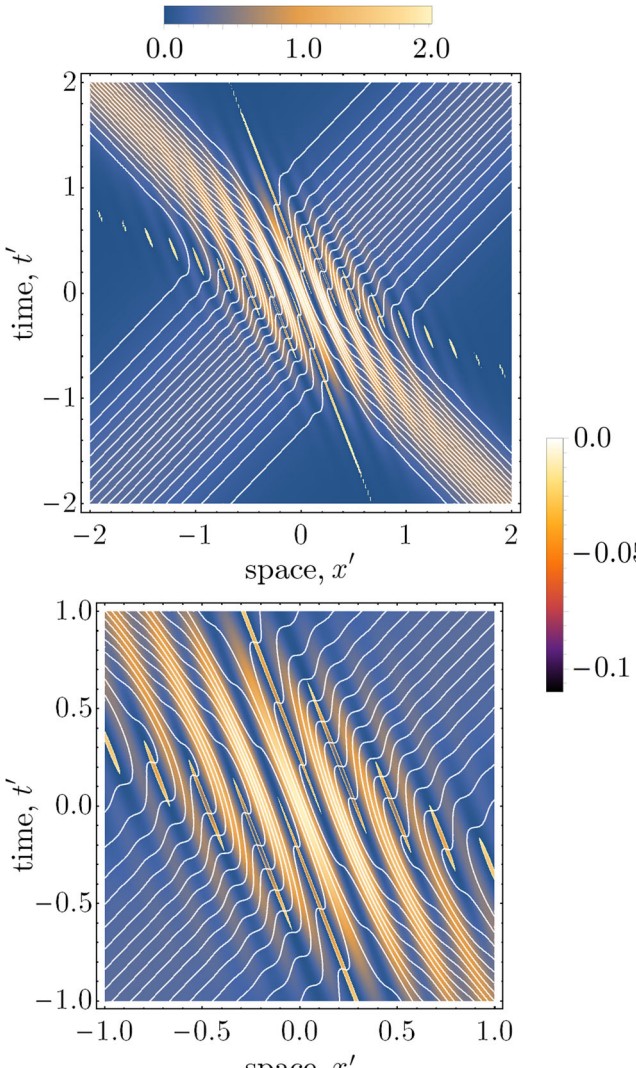

**Fig. 6 Bohmian trajectories in the head on limit, viewed from a boosted reference frame with $v = 0.4$, $k_0/\sigma = 15$, and $\alpha = 1/2$.** The trajectories are superimposed over the corresponding Klein–Gordon density, where the respective legends show the magnitude of the positive and negative density regions. The backwards-in-time trajectories are consistent with our interpretation of the Bohmian velocity field as a coordinate velocity constructed via weak measurements in a particular reference frame. The lower panel shows a zoomed-in view of the trajectories.

photons was reconstructed by performing weak measurements of momentum, postselected at a particular spacetime position. This is complementary to Wiseman's definition, Eq. (26). To perform the weak measurements, the photon polarisation was utilised as a pointer which coupled to the momentum[22]. In our model, constructing the velocity field of the transversely fast photons also requires weak measurements of momentum, so the above-mentioned approach may be applied analogously. Of course, since the transverse velocities are now relativistic, these measurements may be technically challenging but nevertheless possible in principle. Meanwhile, weak measurements of the particle energy in the case of photons essentially require a measurement of its frequency. This has been achieved for example by weakly perturbing the path of a frequency-modulated beam using an adjustable prism, from which weak measurements of the induced frequency shift can be inferred[45–47].

The approach we have introduced in this article motivates numerous pathways for further research. A natural extension of our model would be to study a system with multiple particles, for which the nonlocality of Bohmian mechanics in the relativistic regime may be properly understood. As in the single-particle case, Bohmian-mechanical models for multiple particle scenarios have been developed in the nonrelativistic regime. Braverman and Simon[48] first proposed a model for path-entangled photons in a double-slit experiment, wherein the velocity of the photon entering the apparatus depends on the value of a phase shift applied to the second photon in a spacelike separated region. This was then implemented in the experiment by Mahler et al.[23], giving strong empirical verification of the nature of nonlocal influences in the Bohmian paradigm. How such entanglement manifests in the relativistic domain remains an interesting question. Likewise, it would be worthwhile understanding how our interpretation of the 'guiding metric' might be generalised for multiple particles.

Another important question to answer is the consistency of our model outside the self-imposed optical limit. We studied this regime to obtain physically interpretable results, in line with the constraints of our single-particle theory. The possibility of constructing a full Bohmian quantum field theory, allowing for the study of particle production effects and subcycle optical regimes, remains a tantalising prospect.

## Methods

**Derivation of the relativistic velocity equation.** Recall our proposed form for the weak value definition of the velocity equation:

$$V(x,t) = \frac{\langle x| \langle \hat{k}_w \rangle_{|\psi(t)\rangle}}{\langle x| \langle \hat{H}_w \rangle_{|\psi(t)\rangle}} = \frac{\mathrm{Re}\langle x|\hat{k}|\psi(t)\rangle}{\mathrm{Re}\langle x|\hat{H}|\psi(t)\rangle}. \tag{32}$$

We calculate the numerator and denominator individually. To this end, we need to give values for the $\langle x|\hat{A}|\psi(t)\rangle$ expressions in the theory we use. With $\hat{A} = I$ the identity operator, the identification is trivially $\langle x|\psi(t)\rangle = \psi(x,t)$, the one particle Klein–Gordon field solution. The momentum operator $\hat{k}$ is formed from the generator of displacements in position. So by the standard definition, we can write

$$\langle x|\hat{k}|\psi(t)\rangle = -i\partial_x \psi(x,t) = -i\partial^x \psi(x,t),$$

where the Minkowski metric has been used to raise the derivative index. Similarly, the generator of temporal displacements in the field are associated with the Hamiltonian operator, and hence

$$\langle x|\hat{H}|\psi(t)\rangle = i\partial_t \psi(x,t) = -i\partial^t \psi(x,t).$$

The weak value expressions can be evaluated using these identifications for momentum

$$\begin{aligned}
\langle x| \langle \hat{k}_w \rangle_{|\psi(t)\rangle} &= \mathrm{Re}\frac{\langle x|\hat{k}|\psi(t)\rangle}{\langle x|\psi(t)\rangle}, \\
&= \mathrm{Re}\frac{\langle \psi(t)|x\rangle}{|\langle x|\psi(t)\rangle|^2}\langle x|\hat{k}|\psi(t)\rangle, \\
&= \mathrm{Re}\frac{(-i)\psi^\star(x,t)\partial^x \psi(x,t)}{|\psi(x,t)|^2},
\end{aligned} \tag{33}$$

and energy

$$\begin{aligned}
\langle x| \langle \hat{H}_w \rangle_{|\psi(t)\rangle} &= \mathrm{Re}\frac{\langle x|\hat{H}|\psi(t)\rangle}{\langle x|\psi(t)\rangle}, \\
&= \mathrm{Re}\frac{\langle \psi(t)|x\rangle}{|\langle x|\psi(t)\rangle|^2}\langle x|\hat{H}|\psi(t)\rangle, \\
&= \mathrm{Re}\frac{(-i)\psi^\star(x,t)\partial^t \psi(x,t)}{|\psi(x,t)|^2}.
\end{aligned} \tag{34}$$

Dividing the two weak values yields,

$$\begin{aligned}
V(x,t) &= \frac{\mathrm{Re}(-i)\psi^\star(x,t)\partial^x \psi(x,t)}{\mathrm{Re}(-i)\psi^\star(x,t)\partial^t \psi(x,t)}, \\
&= \frac{2\,\mathrm{Im}\,\psi^\star(x,t)\partial^x \psi(x,t)}{2\,\mathrm{Im}\,\psi^\star(x,t)\partial^t \psi(x,t)}.
\end{aligned} \tag{35}$$

The numerator and denominator are exactly the Klein–Gordon conserved probability current and density, as shown in the main text.

**Position space wavefunction in the head-on limit**. In the head-on limit, the particle energy can be approximated as $E(k) = |k|$. The position space wavefunction is given by

$$\psi_K(x,t) = \sqrt{\alpha} \int dk f_R(k) e^{-i|k|t + ikx}$$
$$+ \sqrt{1-\alpha} \int dk f_L(k) e^{-i|k|t + ikx}. \tag{36}$$

We make the assumption that the wavepackets $f_L(k)$ and $f_R(k)$ are only non-negligible in the regions $k < 0$ and $k > 0$ respectively. This is to remain consistent with the quantum-optical regime that we are restricting our analysis to. One can write

$$\psi_K(x,t) \simeq \sqrt{\alpha} \int_0^\infty dk f_R(k) e^{-i|k|t+ikx}$$
$$+ \sqrt{1-\alpha} \int_{-\infty}^0 dk f_L(k) e^{-i|k|t+ikx}. \tag{37}$$

Using the strictly positive domains of the integral bounds to eliminate the absolute value signs in the exponents yields,

$$\psi_K(x,t) = \sqrt{\alpha} \int_0^\infty dk f_R(k) e^{-ik(t-x)}$$
$$+ \sqrt{1-\alpha} \int_0^\infty dk f_L(-k) e^{-ik(t+x)}. \tag{38}$$

Using the assumptions on $f_L(k)$ and $f_R(k)$, we can then extend the bounds on the integrals without significant error:

$$\psi_K(x,t) = \sqrt{\alpha} \int_{-\infty}^\infty dk f_R(k) e^{-ik(t-x)}$$
$$+ \sqrt{1-\alpha} \int_{-\infty}^\infty dk f_L(-k) e^{-ik(t+x)}. \tag{39}$$

Equation (39) can be evaluated analytically. In the simple case where $k_{0R} = k_{0L} \equiv k_0$ and $\sigma_R = \sigma_L \equiv \sigma$, the wavepacket normalisation constants are simply $(2\pi\sigma)^{-1/4}$, giving

$$\psi_K(x,t) = \mathcal{J}_0[\sqrt{\alpha} \exp[-(t-x)\mathcal{V}_R]$$
$$+ \sqrt{1-\alpha} \exp[-(t+x)\mathcal{V}_L]] \tag{40}$$

where $\mathcal{J}_0 = (2\sigma^2/\pi)^{1/4}$ and

$$\mathcal{V}_R = ik_0 + (t-x)\sigma^2, \tag{41}$$

$$\mathcal{V}_L = ik_0 + (t+x)\sigma^2. \tag{42}$$

Inserting Eq. (40) into the expressions for the relativistic Klein–Gordon probability current and density,

$$j_K(x,t) = 2\,\text{Im}\,\psi^\star(x,t)\partial^x \psi(x,t), \tag{43}$$

$$\rho_K(x,t) = 2\,\text{Im}\,\psi^\star(x,t)\partial^t \psi(x,t), \tag{44}$$

then we straightforwardly obtain Eq. (19) and Eq. (20) stated in the main text.

**Paraxial limit of the relativistic velocity equation**. We can obtain the non-relativistic velocity equation by taking the paraxial limit of our relativistic velocity equation. Firstly, in the paraxial limit, the energy of the photon is $E(k) \simeq k_z + k^2/2k_z$ for $k \ll k_z$. The wavefunction can be written as

$$\psi_S(x,t) = \int dk\, e^{ikx} e^{-ik_z t - \frac{ik^2 t}{2k_z}} f(k) \tag{45}$$

where $f(k) = \sqrt{\alpha} f_R(k) + \sqrt{1-\alpha} f_L(k)$ using $f_L$ and $f_R$ defined in Eqs. (17) and (16). We are also assuming that $f(k)$ is not supported outside of the $k \ll k_z$ approximation. Inserting $\psi_S(x,t)$ into the Klein–Gordon expression for the probability density will give the density appropriate for these approximations, which can be manipulated to give

$$\rho_K(x,t) = 2\,\text{Im} \int dk \int dk' f^\star(k) f(k')$$
$$\exp[-ik'x] \exp\left[ik_z t + \frac{itk'^2}{2k_z}\right]$$
$$\left(-\frac{\partial}{\partial t}\right) \left(\exp[ik'x] \exp\left[-ik_z t - \frac{itk'^2}{2k_z}\right]\right),$$
$$= 2\,\text{Im} \int dk \int dk' f^\star(k) f(k') \left(ik_z + \frac{ik'^2}{2k_z}\right)$$
$$\exp[-i(k-k')x] \exp\left[\frac{it}{2k_z}(k^2 - k'^2)\right],$$

Recognising that $k'^2$ can be expressed in differential operator form as $-\partial^2/\partial x^2$ and that the double integral expression is simply $\psi_S^\star(x,t)\psi_S(x,t)$, we obtain the

simplified form

$$\rho_K(x,t) = 2\,\text{Im}\left[ik_z|\psi_S(x,t)|^2 - \frac{i}{2k_z}\psi_S^\star(x,t)\frac{\partial^2 \psi_S(x,t)}{\partial^2 x}\right], \tag{46}$$
$$\simeq 2k_z|\psi_S(x,t)|^2 = 2k_z\rho_S(x,t),$$

where in the final line the approximation used is that the double spatial derivative of $\psi_S$ is negligible compared to $k_z^2$. The velocity equation in the paraxial limit thus becomes

$$V(x,t) = \frac{1}{2k_z} \frac{2\,\text{Im}\,\psi_S^\star(x,t)\partial^x \psi_S(x,t)}{|\psi_S(x,t)|^2}. \tag{47}$$

This is simply the nonrelativistic definition of the velocity, expressed in terms of the Schrödinger probability current and density. This is exactly the form obtained using Wiseman's nonrelativistic weak value formula in ref. [20].

**Analytic expression for the Bohmian velocity in the paraxial limit**. Let us derive the analytic form of the paraxial limit velocity. Again, applying the optical approximation used in Eq. (37), for an initial superposition of left- and right-moving wavepacket the wavefunction in the paraxial limit is given by

$$\psi_S(x,t) = \sqrt{\alpha} e^{-ik_z t} \int dk\, e^{ikx - \frac{ik^2 t}{2k_z}} f_R(k)$$
$$+ \sqrt{1-\alpha} e^{-ik_z t} \int dk\, e^{-ikx - \frac{ik^2 t}{2k_z}} f_L(-k). \tag{48}$$

The integrals can be evaluated analytically. The components of the probability current can be split into left- and right-moving parts, and an interference term which mixes these components. These are given by

$$j_L(x,t) = -(1-\alpha)\mathcal{C}_0 \exp\left[-\frac{2(k_0 t + k_z x)^2 \sigma^2}{k_z^2 + 4t^2\sigma^4}\right],$$
$$j_R(x,t) = \alpha\mathcal{D}_0 \exp\left[-\frac{2(k_0 t - k_z x)^2 \sigma^2}{k_z^2 + 4t^2\sigma^4}\right], \tag{49}$$
$$j_I(x,t) = \sqrt{\alpha(1-\alpha)}\mathcal{L}_0 \exp\left[-\frac{2k_{tx}\sigma^2}{k_z^2 + 4t^2\sigma^4}\right],$$

where we have defined $k_{tx} = k_0^2 t^2 + k_z^2 x^2$ and

$$\mathcal{C}_0 = \mathcal{M}_0(k_0 k_z - 4tx\sigma^4),$$
$$\mathcal{D}_0 = \mathcal{M}_0(k_0 k_z + 4tx\sigma^4),$$
$$\mathcal{M}_0 = \sqrt{\frac{2}{\pi}} \frac{k_z\sigma^2}{(k_z^2 + 4t^2\sigma^4)^{3/2}}, \tag{50}$$

and

$$\mathcal{L}_0 = 4\sigma^2 t \mathcal{M}_0 \left[2x\sigma^2 \cos\left[\frac{2k_0 k_z^2 x}{k_z^2 + 4t^2\sigma^4}\right]\right.$$
$$\left. + k_0 \sin\left[\frac{2k_0 k_z^2 x}{k_z^2 + 4t^2\sigma^4}\right]\right]. \tag{51}$$

Similarly, the density can likewise be decomposed into left- and right-moving parts, and an interference term. These are,

$$\rho_L(x,t) = (1-\alpha)\mathcal{K}_0 \exp\left[-\frac{2(k_0 t + k_z x)^2 \sigma^2}{k_z^2 + 4t^2\sigma^4}\right],$$
$$\rho_R(x,t) = \alpha\mathcal{K}_0 \exp\left[-\frac{2(k_0 t - k_z x)^2 \sigma^2}{k_z^2 + 4t^2\sigma^4}\right], \tag{52}$$
$$\rho_I(x,t) = \sqrt{\alpha(1-\alpha)}\mathcal{K}_e \exp\left[-\frac{2k_{tx}\sigma^2}{k_z^2 + 4t^2\sigma^4}\right],$$

where we have defined

$$\mathcal{K}_0 = \sqrt{\frac{2}{\pi}} \frac{k_z\sigma^2}{\sqrt{k_z^2 + 4t^2\sigma^4}},$$
$$\mathcal{K}_e = 2\mathcal{K}_0 \cos\left[\frac{2k_0 k_z^2 x}{k_z^2 + 4t^2\sigma^4}\right]. \tag{53}$$

These expressions for the probability and current density are used to construct the trajectories shown in Fig. 5.

**Obtaining the reference frame in which $\rho(x,t)$ is positive definite**. It was argued in the discussion that under the optics approximation, there always exists a global coordinate transformation which ensures that the probability density $\rho(x,t)$ is positive-definite for all $(x,t)$. This was shown straightforwardly when the

left- and right-moving components of $\psi(x,t)$ possessed equally weighted centre frequencies ($k_0$) and bandwidths ($\sigma$). The only other scenario of interest occurs when the left- and right-moving components are unequal. The probability density in such a scenario is given by

$$\rho(x,t) = \mu^2 + \nu^2 + 2\mu\nu\mathcal{F}_0 \tag{54}$$

where

$$
\begin{aligned}
\mu &= \sqrt{\alpha}\sqrt{\sigma_R}\left(\frac{2}{\pi}\right)^{1/4}\exp\left[-(t-x)^2\sigma_R^2\right], \\
\nu &= \sqrt{1-\alpha}\sqrt{\sigma_L}\left(\frac{2}{\pi}\right)^{1/4}\exp\left[-(t+x)^2\sigma_L^2\right], \\
\mathcal{F}_0 &= \frac{k_{0R}+k_{0L}}{2\sqrt{k_{0R}k_{0L}}}\left[\cos\left[k_{0L}V - k_{0R}U\right] - \left[V\sigma_L^2 - U\sigma_R^2\right]\sin\left[k_{0L}V - k_{0R}U\right]\right],
\end{aligned}
\tag{55}
$$

where we have expressed the spacetime variables in terms of the light-cone coordinates $V = (t+x)$ and $U = (t-x)$, $(k_{0R}, \sigma_R)$, $(k_{0L}, \sigma_L)$ denotes the right- and left-moving parameters respectively and we have included the relativistic normalisation factor for the left- and right-moving parts of the wavefunction. A similar expression exists for $j(x,t)$. Such a scenario mimics one in which the wavepacket frequencies/bandwidths are Doppler shifted (i.e. undergo a boost), and thus there may arise points where $\rho(x,t) < 0$ is true.

Now, consider Eq. (54) from a boosted reference frame with velocity $v$, yielding

$$\rho'(x',t') = \mu'^2 + \nu'^2 + 2\mu'\nu'\mathcal{F}'_0, \tag{56}$$

where

$$
\begin{aligned}
\mu' &= \sqrt{\alpha}\sqrt{\sigma'_R}\exp\left[-(t'-x')^2\sigma_R'^2\right], \\
\nu' &= \sqrt{1-\alpha}\sqrt{\sigma'_L}\exp\left[-(t'+x')^2\sigma_L'^2\right], \\
\mathcal{F}'_0 &= \frac{k'_{0R}+k'_{0R}}{2\sqrt{k'_{0R}k'_{0L}}}\left[\cos\left[k'_{0L}V' - k'_{0R}U'\right]\right. \\
&\qquad \left. - \left[V'\sigma_L'^2 - U'\sigma_R'^2\right]\sin\left[k'V' - k'U'\right]\right],
\end{aligned}
\tag{57}
$$

and we have defined

$$k'_{0R} = \sqrt{\frac{1-v}{1+v}}k_{0R}, \qquad \sigma'_R = \sqrt{\frac{1-v}{1+v}}\sigma_R, \tag{58}$$

$$k'_{0L} = \sqrt{\frac{1+v}{1-v}}k_{0L}, \qquad \sigma'_L = \sqrt{\frac{1+v}{1-v}}\sigma_L, \tag{59}$$

and likewise $V' = (t'+x')$ and $U' = (t'-x')$. We make two key observations. Firstly, there is always a choice of $v$ which causes either the $k_0$'s to be equal, or the $\sigma$'s to be equal. For example, by choosing

$$v = \frac{k_{0R} - k_{0L}}{k_{0R} + k_{0L}} \tag{60}$$

then

$$k'_0 \equiv k' = k'_{0L} = \sqrt{k_{0R}k_{0L}}. \tag{61}$$

Secondly, if the optics approximation is satisfied in the original frame, it is satisfied in all other reference frames since $(k'_{0R}, \sigma'_R)$ and $(k'_{0L}, \sigma'_L)$ for the left- and right-moving components scale identically, Eqs. (58) and (59). That is, if $k_{0R} \gg \sigma_R$ and $k_{0L} \gg \sigma_L$, then $k'_{0R} \gg \sigma'_R$ and $k'_{0L} \gg \sigma'_L$. Let us now consider the density after having made the aforementioned boost, Eq. (60):

$$\rho'(x',t') = \mu'^2 + \nu'^2 + 2\mu'\nu'\mathcal{Y}_0 \tag{62}$$

where

$$\mathcal{Y}_0 = \cos\left(2k'_0 x'\right) - \frac{V'\sigma_L'^2 - U'\sigma_R'^2}{k'_0}\sin\left(2k'_0 x'\right). \tag{63}$$

In Eq. (62), the regime of interest occurs when

$$\delta_R := U'\sigma'_R \lesssim \mathcal{O}(1), \tag{64}$$

$$\delta_L := V'\sigma'_L \lesssim \mathcal{O}(1) \tag{65}$$

otherwise $\rho'(x',t')$ will be exponentially suppressed and the particle will have zero probability of being found at $(x',t')$. In this limit, Eq. (63) reduces to

$$\mathcal{Y}_0 \simeq \cos(2k'_0 x') \tag{66}$$

where we have invoked the optics approximation, $k'_0 \gg \sigma'_R, \sigma'_L$, making the coefficient of the $\sin(2k'_0 x')$ term negligible. The full probability density in this limit is thus given by

$$\rho'(x',t') \simeq \mu'^2 + \nu'^2 + 2\mu'\nu'\cos(2k'_0 x') \geq 0 \tag{67}$$

which completes the proof.

## Data availability

The data presented in the analysis can be reproduced using code which is available from the corresponding author on reasonable request.

## Code availability

The code utilised for the analysis in the current study is available from the corresponding author on reasonable request.

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

## Acknowledgements

E.A. and T.C.R. acknowledge motivating discussions with Howard Wiseman. A.P.L. acknowledges support from BMBF (QPIC) and the Einstein Research Unit on Quantum Devices. This research was supported by the Australian Research Council Centre of Excellence for Quantum Computation and Communication Technology (Project No. CE170100012).

## Author contributions

J.F., A.P.L., and T.C.R. contributed to all aspects of the research. E.A. contributed to the initial aspects of the paper.

## Competing interests

The authors declare no competing interests.
