## [Peer Review File · Nature Communications]

Relativistic Bohmian trajectories of photons via weak measurementsREVIEWER COMMENTS

Reviewer #1 (Remarks to the Author):

In this paper, the authors propose a relativistic theory of Bohmian trajectories, mathematically based on Klein-Gordon current and operationally based on weak measurements. While this is interesting, I see two problems with this proposal.

First, the idea to base relativistic Bohmian theory on Klein-Gordon current is not new. Second, their specific operational proposal is not relativistic invariant.

Relativistic Bohmian mechanics based on Klein-Gordon current has been discussed extensively in the literature, none of which is mentioned in this paper. Examples include K. Berndl et al (arXiv:quant-ph/9510027) and a long series of papers by

H. Nikolic (most of which are cited in arXiv:1205.1992). In one of those papers it was even suggested that it can be justified by weak measurements (arXiv:hep-th/0702060, paragraph around Eq. (95)). The authors should make a serious research of existing literature and compare their results with those which already exist.

The operational approach they propose is not relativistic invariant because the position eigenstate (7) is not relativistic invariant. That's because the integration measure dk is not relativistic invariant. This problem cannot be solved by taking the relativistic invariant measure $dk/E(k)$, because then the state $|x\rangle$ is no longer localized in space. Those problems are well known in the literature, often summarized by saying that position operator in quantum theory cannot be defined in a relativistic invariant manner. The most famous paper on this topic is T.D. Newton and E.P. Wigner, Rev. Mod. Phys. 21, 400 (1949).

If the authors can revise the paper such that those issues are addressed properly, the paper could be suitable for publication. In that case, I would also suggest one minor correction. The authors say that Eq. (5) is relativistically invariant, but since this quantity does not transform as a Lorentz scalar, it is not strictly true. It's perhaps nitpicking and nothing but semantics, but the authors should be more precise about that.

Reviewer #2 (Remarks to the Author):

The authors propose equations for defining Bohmian trajectories for photons. It would indeed be of interest to see how Bohmian trajectories can be defined for photons. So far, they can be defined only for spin-1/2 and spin-0 particles. However, the proposal in the manuscript is not a convincing achievement, for the following reasons:

- 1) The wave function is taken to be scalar-valued and thus to correspond to spin 0, whereas photons have spin 1. So, the proposal is unphysical.
- 2) Despite the authors' claim, the proposed law of motion, given by (10) and (11), is not new but coincides with equations (11.2) and (11.3) on page 233 in [6], published in 1993.
- 3) Among experts, this law of motion is mostly regarded as physically unconvincing because the resulting world lines can go back and forth in time as the current vector field (11) sometimes points towards the future and sometimes towards the past.
- 4) The authors propose to avoid the past-pointing current by assuming a "distribution of wavevectors [...] concentrated well away from zero" (p. 2). However, an equation that makes sense only in a special regime appears much less convincing or interesting as a candidate for a

physical law.

By the way, the authors claim that their equation of motion is the first relativistic one; this is not correct, as Bohm proposed a relativistic equation of motion on the basis of a Dirac wave function in Prog. Theor. Phys. 9: 273 (1953), also described in Chap. 12 of [6].

I do not see anything substantial added to the literature by the manuscript. I recommend rejection.

Reviewer #3 (Remarks to the Author):

The central question to this work is the introduction of a relativistic expression for the Bohmian guidance condition, from which trajectories describing the evolution of the quantum system can be synthesized after integration. More specifically, what makes appealing the equation determined is the fact that it would allow the inference of Bohmian trajectories for photons in the relativistic regime by performing weak measurements of both momentum and energy, as it has already been done by Steinberg's group in Toronto in the nonrelativistic regime. Furthermore, a simple model to extend this equation to the domain of general relativity is also suggested by considering a modified Alcubierre metric, although this part is only slightly elaborated.

The idea is interesting, because, to the best of my knowledge, there have not been proposals to tackle the issue of experimentally determining (even though only at an inferential level, as in the case of the experiments mentioned before) Bohmian-type quantum trajectories in the relativistic domain making use of weak measurements. The chosen system to proceed, as I understand, would be photons, which increases the interest of the proposal, because the weak-measurement techniques are quite developed for photons.

However, in my opinion, there are some questions or aspects that need first to be properly clarified and/or conveniently revised, which might contribute to improve the context, understanding and potential reach and interest of the work:

* At the conceptual level, although Bohmian mechanics is widely regarded as a deterministic, nonlocal theory, as it is also taken for granted here, note that it is nothing but another reformulation of quantum mechanics. As Bohm himself acknowledges, what he did was to rationalize what a quarter of a century earlier de Broglie had only postulated, namely that quantum systems can be understood as point-like particles (like any classical system) acted by a wave field (making here a capital difference with classical systems), apart from eventual external fields. There is still an on-going debate on whether both approaches are the same or, on the contrary, there are some divergences, but this is a minor issue to the discussion here (actually, totally irrelevant). This idea came up after Bohm's attempts to disprove the conclusion of von Neumann's theorem on the impossibility of a hidden-variable-based explanation of quantum phenomena, as the authors might know well.

So far, so good, except for the fact that, alternatively to de Broglie, the same year that Schrödinger published his equation, Madelung sent to publication a manuscript with a reformulation of the former's equation in a hydrodynamic form [Z. Phys. 40, 322 (1927)]. By means of a polar ansatz identical to the one later on considered by Bohm, but directly associating it with a system energy (to some extent, in the Hamiltonian way), he concludes how the Schrödinger equation should be reinterpreted in hydrodynamic terms, emphasizing towards the end, for instance, that the so-called quantum jumps should be understood in a continuous rather than discrete manner. Some other conclusions linking the probability density with ensembles of identical realizations are also worth noting, though.

Accordingly, it seems there is nothing new in Bohmian mechanics as a theory, as widely claimed in

the literature (except by Bohm himself, who indeed disliked the term "Bohmian mechanics"), but just another formulation of the standard quantum mechanics, although less used because of unjustified prejudices, particularly, the fact that it incorporates the notion of well-defined trajectory. But there is nothing wrong about this, for in the same way that hydrodynamics includes streamlines that serve to monitor the flow of a fluid, the so-called Bohmian trajectories are streamlines that serve to monitor the flow of the probability density.

In sum, although the manuscript tries to, somehow, put the emphasis on the old idea of hidden variables and alternative quantum theories, this is nothing but introducing more of unnecessary noise in the literature on the issue.

* In connection to the previous point it should also be stressed that, contrary to the authors' claim, quantum (Bohmian) trajectories have not been observed experimentally, but they have only been experimentally inferred. This is a totally different question, which has induced to misconceptions in the literature and related debates at conferences regarding Steinberg's experiments. In such experiments, with the aid of both weak and strong (von Neumann) measurements the transverse flux and the probability density are respectively determined. From those experimental data, "taking for granted" a Bohmian-based guidance equation, the corresponding trajectories were inferred. However, such trajectories do not represent the paths followed by the photons themselves, but only streamlines along which the electromagnetic energy flows. This is something somehow acknowledged in the title of Steinberg's Science paper, Ref. 18, by denoting them as "average trajectories", which takes us back again to Madelung's viewpoint. Note that there is no empirical evidence that a single photon can be uniquely associated with a given trajectory, because weak measurements determine, in the end, average values, even though in a particular manner.

Furthermore, it is worth mentioning that, contrary to a general believe, the possibility to measure other quantities than the usual expectation values, based on the notion of von Neumann's strong measurement, is already incipient in Hirschfelder's equations of change, suggested by the late 1970s [J. Chem. Phys. 68, 5151 (1978)]. These equations are intended to measure, for instance, the flow of energy in the system from one part to another one, which is actually what Bohmian trajectories provide us with in the case of photons (electromagnetic fields).

* Regarding the approach considered here, what calls my attention is the fact that the authors wish to determine photon trajectories (understand this in line with the above comments), but they make use of the Klein-Gordon equation, which is only applicable to scalar (spinless) particles. It is clear that the framework it provides helps to "capture the essential Lorentz relativistic properties of photons", as it is acknowledged by the authors. However, photons require a more complete treatment, such as the Dirac equation (a QED extension of Maxwell's equations), at least like the one provided about twenty years earlier by Ghose and coworkers (Ref. 23), based on Dirac's equation. Other important works developed by Dewdney and Horton or Nikolic on relativistic extensions of Bohmian mechanics, published along the 2000s, seem also to have been disregarded, although they constitute an appropriate theoretical ground.

Some of those works, though, directly attack the Klein-Gordon equation and provide a proper guidance equation [Found. Phys. 32, 463 (2002); J. Phys. A 33, 7337 (2000); Found. Phys. Lett. 18, 549 (2005)]. Same happens with the more recent work by Capistrano, cited in Ref. 24 (please, remove Jalalzadeh, since he finally did not coauthorized the published version of the paper). However, it seems none of them have been seriously considered as a starting point (even though with the aforementioned limitations for photons).

* The above point poses another important issue: although Bohm introduces the so-called guidance equation as a postulate, this is not an essential requirement to introduce (Bohmian) trajectories, for such an expression directly and naturally arises from the fact that a transport equation rules the diffusion of the probability density in configuration space, namely the continuity equation. Accordingly, the quantum flux is directly related with the probability density by means of

a velocity field. This basic relation also appears in other wave theories, which enables the extension of the same ideas to such theories (or, in other words, to extrapolate notions from those other theories to the quantum realm, essentially what Madelung did). This is an important issue, because it directly implies that, whenever we are able to find a transport equation in quantum mechanics, we can describe the phenomenon in a hydrodynamic fashion. Accordingly, there must be a unique relationship between a density function and the corresponding flux.

Although it is mentioned that the flux and probability density here satisfy a continuity equation, namely Eq. (12), it is not clear where does Eq. (6) exactly come from? If one goes to Sec. IIIA, "Methods", it seems that Eq. (32) [the same as Eq. (6)] does not arise as a direct consequence of a transport process, but it is "proposed". This point should be properly clarified. Otherwise, it constitutes an important break from the ideas underlying Bohmian mechanics (some of them have already been slightly sketched in the comments above), entering the realm of speculation.

* On another note, it is found that Eq. (6) is given in terms of momentum divided by energy. This sounds odd to me, particularly in the case of light, where one would expect energy (or frequency) over momentum (or wavevector), which has the dimensions of a velocity (actually, in vacuum, we should obtain something proportional to c) both in quantum mechanics and also in special relativity.

In nonrelativistic Bohmian mechanics the r.h.s. of the guidance equation is directly related with the real part of the local evaluation (in space) of the usual momentum operator. Here, I am afraid that I might be missing something, so I would acknowledge a clarification.

Furthermore, should not the velocity be defined, in general, in terms of a proper time rather than the usual space and time coordinates? It seems that here time has also been used as an evolution parameter instead of a variable itself, such as the spatial coordinate, thus making little distinction with nonrelativistic quantum mechanics.

* Although in the relativistic domain, the analysis of head-on collisions is interestingly similar to the one published earlier on by Sand and Miret-Artés about wave-packet head-on collisions [J. Phys. A 41, 435303 (2008)], including the α parameter specifying the contribution from each side, which is not included here in the reference list. Independently of this, it is interesting to note that the differences with that work seem to be only the origin of time, for the trajectories (and the behavior of the probability densities) look pretty similar to the results expected in a nonrelativistic scenario, like the one considered by Sand and Miret-Artés. Actually, in the latter case, a simple translation in time would render similar figures. A direct comparison cannot be set, because the density is omitted in the referred work, but the behavior of the trajectories is similar.

RESPONSE TO REVIEWERS

Referee 1

- Comment:

In this paper, the authors propose a relativistic theory of Bohmian trajectories, mathematically based on Klein-Gordon current and operationally based on weak measurements. While this is interesting, I see two problems with this proposal.

Response:

We thank the referee for taking interest in our paper and proposal. We would like to clarify from the outset that our relativistic theory is *not based* on the Klein-Gordon current. The formal mathematical objects which emerge (e.g. the Klein-Gordon conserved quantities) must be understood as arising *incidentally* from our operational starting point, that is the theory of weak measurements. This operational *measurement* formalism must be the lens through which our results are interpreted. Understanding the theory of weak measurements and weak values as the foundational starting point for our paper should clarify the referee's two concerns mentioned below.

• Comment:

First, the idea to base relativistic Bohmian theory on Klein-Gordon current is not new. Second, their specific operational proposal is not relativistic invariant. Relativistic Bohmian mechanics based on Klein-Gordon current has been discussed extensively in the literature, none of which is mentioned in this paper. Examples include K. Berndl et al (arXiv:quant-ph/9510027) and a long series of papers by H. Nikolic (most of which are cited in arXiv:1205.1992). In one of those papers it was even suggested that it can be justified by weak measurements (arXiv:hep-th/0702060, paragraph around Eq. (95)). The authors should make a serious research of existing literature and compare their results with those which already exist.

Response:

As emphasised above, we do not base our relativistic Bohmian theory on the Klein-Gordon conserved current/field. Our approach is a relativistic reformulation of Wiseman's original work (NJP (9) 165 (2007)), taking as a starting point the construction of the relativistic velocity field from weak measurements. Our results must be interpreted from this operational perspective, and are thus quite distinct from the mentioned works. To address this concern which is common to each of the referees, we have significantly reordered the introductory paragraphs to reflect our emphasis that the novelty of our construction comes from this weak-measurement foundation. Similar changes in emphasis have been made on page 3 and 8 of the resubmitted manuscript. We have also revised the title of our paper to be '*Relativistic Bohmian trajectories of photons via weak measurements*' to reflect our intended emphases.

Beginning with our proposed weak measurement formalism, we then evaluate the weak values using the scalar Klein-Gordon theory, resulting in the appearance of the conserved current components. This is significant because these quantities inherit the relativistic properties of the Klein-Gordon field. However we emphasise that such a connection is only drawn after defining the Bohmian velocity field via weak measurements of momentum and energy.

We thank the referee for pointing out the previous works to us and have added them to the reference list where appropriate. We still affirm that our paper represents a significant advance over these prior investigations.

- The paper by K. Berndl et al (arXiv:quant-ph/9510027) only mentions the Klein-Gordon theory in its introduction. The crux of the paper revolves around formulating the probability density $\Psi^\dagger\Psi$ in a Lorentz invariant way. We have included a reference to the paper in the introduction (page 2).
- The paper by H. Nikolic (arXiv:1205.1992) takes the conventional route in Bohmian mechanics by writing the wavefunction in polar form and arriving

at a velocity field. However it is disconnected from any measurement model. We have included a reference to this paper in the introduction (page 2), as well as the additional papers by Nikolic: Found. Phys. Lett. 17, 363–380 (2004), Found. Phys. Lett. 18, 123–138 (2005). Found. Phys. Lett. 18, 549–561 (2005), which are helpful as historical links for the reader.

- The paper by H. Nikolic (arXiv:hep-th/0702060) references Wiseman’s 2007 paper once, and suggests that a relativistic extension to his theory could be made (without suggesting how that might be achieved). Our paper discusses how a naïve relativistic extension to Wiseman’s original paper does *not* work. We then show the correct way to construct the relativistic velocity field from weak measurements. Neither of these have been shown previously. We have included this in the reference list of our paper.

- Comment:

The operational approach they propose is not relativistic invariant because the position eigenstate (7) is not relativistic invariant. That’s because the integration measure dk is not relativistic invariant. This problem cannot be solved by taking the relativistic invariant measure $dk/E(k)$, because then the state x) is no longer localized in space. Those problems are well known in the literature, often summarized by saying that position operator in quantum theory cannot be defined in a relativistic invariant manner. The most famous paper on this topic is T.D. Newton and E.P. Wigner, Rev. Mod. Phys. 21, 400 (1949).

Response:

We maintain that our approach is by construction relativistic. At various points in the paper (including the title) we used the language ‘relativistically invariant’, which may not have conveyed our intended meaning. What was intended was that beginning with Eq. (5) and connecting this with weak measurements and the Klein-Gordon current, the velocity field transforms as would be expected for a velocity under a Lorentz transformation i.e. in a relativistic way. The velocity field of Eq. (5) is the coordinate velocity constructed from derivatives with respect to coordinates for the components of the relativistic two-vector $x = (t, \mathbf{x})$. Transforming into a Lorentz-boosted frame is a matter of appropriately transforming the relevant components of the spacetime two-vector (t, \mathbf{x}) and the conserved current two-vector (ρ, \mathbf{j}) using standard Lorentz transformations.

As the referee rightly notes, the problem with the position eigenstate x) is well-documented in the literature. The key point here is that such a problem is avoided by defining $V(x, t)$ in terms of an in-principle measurement in *a particular reference frame*. When the operational weak measurement approach is used in a given

reference frame, one can then consistently apply the Lorentz transformation to obtain the velocity field in different reference frames. Conversely, if we move into a different frame and apply our operational definition, we can relate the velocities we obtain in this frame to those that would be obtained in the original frame through a standard Lorentz transformation. At no point does the operational approach require the transformation of position eigenstates $|x\rangle$ between frames.

• Comment:

If the authors can revise the paper such that those issues are addressed properly, the paper could be suitable for publication. In that case, I would also suggest one minor correction. The authors say that Eq. (5) is relativistically invariant, but since this quantity does not transform as a Lorentz scalar, it is not strictly true. It's perhaps nitpicking and nothing but semantics, but the authors should be more precise about that.

Response:

Thank you for the helpful comments in clarifying aspects of our paper which have helped us to communicate our results more clearly; we believe we have addressed the concerns raised above. Regarding the suggested correction: this is related to the discussion on page 4 of this letter, and as discussed we have clarified the language to read 'Lorentz covariant' throughout the paper.

Referee 2

• Comment:

The authors propose equations for defining Bohmian trajectories for photons. It would indeed be of interest to see how Bohmian trajectories can be defined for photons. So far, they can be defined only for spin-1/2 and spin-0 particles.

Response:

Thank you for recognising the potential interest of our paper among the scientific community.

• Comment:

However, the proposal in the manuscript is not a convincing achievement, for the following reasons:

1) The wave function is taken to be scalar-valued and thus to correspond to spin 0, whereas photons have spin 1. So, the proposal is unphysical.

Response:

Firstly, and as stated in our initial summary at the beginning of this letter: a description of particles with spin would be an extension to the current work (i.e. the

promotion of the scalar wavefunction to a vector wavefunction). We do not consider the use of a scalar-valued wavefunction as a limitation of our approach, as it is reasonable to consider the simplest and most idealised model when exploring novel physical concepts. Indeed, scalar fields are used commonly in relativistic quantum information, quantum mechanics, and quantum field theory as good approximations to, and the simplest way of describing photons and the light-matter interaction. Referee 1 also raised the paper by H. Nikolic (arXiv:1205.1992) which reviews the necessary elements of the relativistic spin-0, spin-1/2, and spin-1 wave equations in the context of Bohmian mechanics. There (and in the research fields stated above), the scalar Klein-Gordon theory is commonly used so as to highlight the property of Lorentz covariance and is a convenient simplification of the vector spin-1 theory. The spin-1 theory can be constructed by using the scalar theory for vector components with constraints between them; but any of the individual components are exactly obtained from the scalar Klein-Gordon equation.

Secondly, adding vector components to the field does not add considerable insight into the relativistic properties of the trajectories. As emphasised in the main text of our paper, the scalar-valued framework ‘captures the essential Lorentz relativistic properties of photons’ which was our primary interest.

Finally, in the relativistic regime we consider, the x and z dynamics are decoupled (i.e. the modes are plane waves in the z -direction). In this scenario, the polarisation degrees of freedom are decoupled and therefore treating the wavefunction as a scalar is an appropriate simplification. There is no fundamental change when introducing couplings between the polarisations. As mentioned, if one wanted to do this for completeness, the reference by H. Nikolic (arXiv:1205.1992) provided by referee 1 shows that the extension of the spin-0 theory to a full QED theory is straightforward. However, such details would obscure the key message of our paper (the measurement-based formulation of relativistic Bohmian mechanics, the resultant particle trajectories and their interpretation), and would be best left for a future extension.

We have added additional clarifying sentences throughout the paper to address these concerns, for example: top of page 2 right-hand column, discussion around Eq. (9).

• Comment:

2) Despite the authors’ claim, the proposed law of motion, given by (10) and (11), is not new but coincides with equations (11.2) and (11.3) on page 233 in [6], published in 1993.

Response:

Unfortunately, the equations cited by the referee do not match up with our version of Bohm's book. We believe that the referee is referring to Eq. (12.1)-(12.3) which state the conserved current density obtained from the Dirac equation:

$$\mathbf{j} = \psi^\dagger \boldsymbol{\alpha} \psi \quad (1)$$

$$j_0 = \psi^\dagger \psi \quad (2)$$

and the continuity equation and velocity:

$$0 = \frac{\partial j_0}{\partial t} + \nabla \cdot \mathbf{j}, \quad (3)$$

$$\mathbf{v} = \frac{\mathbf{j}}{j_0}. \quad (4)$$

Assuming this is the case, we state in our paper that the density $|\psi|^2$ does not transform relativistically under Lorentz boosts for scalar-valued wavefunctions. Therefore Eq. (4) above is not directly related to our investigation which studies relativistic scalar particles.

Nevertheless, the referee's comment conveys a misunderstanding regarding the main claim of our paper. As emphasised in our response to referee 1, we do not use relativistic quantum mechanics as a starting point (e.g. the Klein-Gordon or Dirac equations). The fundamental novelty of our proposal comes from it being an *operational measurement approach* used to construct the Bohmian velocity field. This reformulation of Wiseman's original paper has not been proposed before. As stated in our response to referee 1 (page 3 of this letter), the connection we make between weak values and the Klein-Gordon conserved quantities is novel but also natural; we begin from a relativistic weak-value equation and likewise obtain a velocity field expressed in terms of relativistic quantities.

In line with this, one should not interpret our paper as attempting to propose a new 'law of motion'. The derived velocity field and trajectories arise from our operational weak measurement model under physical assumptions consistent with the focus on a single-particle theory (this is addressed further on page 9 of this letter).

As stated in our response to referee 1 on page 3 of this letter, we have made changes to the emphasis of our introduction of the weak-value construction throughout the paper, and have revised the title. This is especially reflected in the reordering of the introduction, where we bring in the important background of Wiseman's nonrelativistic weak-value approach.

• Comment:

3) *Among experts, this law of motion is mostly regarded as physically unconvincing because the resulting world lines can go back and forth in time as the current vector field (11) sometimes points towards the future and sometimes towards the past.*

Response:

Firstly, our weak measurement model makes sense of the emergence of backwards-in-time trajectories arising from Lorentz boosts. The velocity field is a coordinate velocity obtained from measurements performed in a particular reference frame (in this case, the static lab frame). In our construction it is always possible to perform a global coordinate transformation where the velocity field is forward directing, such as the backwards-in-time trajectories shown in Fig. 6. The Bohmian velocity field is frame-dependent, and for any theory admitting superluminal velocities (which are not controversial, see for example H. Nikolic (arXiv:1006.1986) cited by referee 1, and our discussion of the Alcubierre metric), there will likewise exist a reference frame in which the current vectors are backwards pointing.

Secondly, the existence of backwards-in-time trajectories induced by transformations of relativistic reference frame must be distinguished from their emergence due to violations of the optics approximation employed in our paper. When $k_0 < \sigma$, the tail of the Gaussian wavepacket $f(k)$ impinges on the negative frequency domain of k -space, which is widely understood as the regime in which subcycle particle production effects occur. Since our proposal is currently a single-particle theory, we were careful to emphasise this difference. On the other hand, one may remain within the optics approximation and still obtain backwards-in-time trajectories through a Lorentz transformation; such trajectories are consistent with our framework.

We finally point out that many experts consider backwards-in-time worldlines as a matter of interpretation (for example, H. Nikolic in Int. J. Mod. Phys. A 25, 1477, cited by referee 1, associates these with particle creation/annihilation processes) rather than a physically unconvincing property. Our weak-value interpretation provides a unique explanation for backwards-in-time trajectories, distinguishing those emerging due to relativistic frame transformations, as compared with particle creation/annihilation processes.

We have clarified in the discussion (page 7 and 8 of resubmitted manuscript) the intended distinction between the two ways in which backwards-in-time trajectories can arise – one consistent, and the other inconsistent with our physical setup. We have likewise added extra sentences in the introduction to emphasise the domain of applicability of our theory (page 2 top of the right-hand column). As mentioned

on page 2 and 3 of this letter, we have included references to the additional works mentioned by Nikolic.

• Comment:

4) *The authors propose to avoid the past-pointing current by assuming a “distribution of wavevectors [...] concentrated well away from zero” (p. 2). However, an equation that makes sense only in a special regime appears much less convincing or interesting as a candidate for a physical law.*

Response:

We do not claim to propose a candidate for a new physical law. Instead, the motivation of our paper is to reformulate and extend Wiseman’s nonrelativistic weak value formalism to relativistic regimes. Our paper states that our motivation was the reinvigoration of interest following his paper and the successful experiments by Kocsis et. al. and Mahler et. al (which were themselves motivated by the weak measurement model) in the nonrelativistic regime. We have studied the kinds of particle trajectories which might arise in a standard quantum optical interferometry setup. As stated above, the assumption that the ‘*distribution of wavevectors ... is concentrated well away from zero*’ was taken to maintain consistency with the domain of applicability of our single-particle theory, also yielding trajectories which might be inferred in an experiment.

This concern is addressed with additional comments on page 2, top of the right-hand column of our resubmitted manuscript.

• Comment:

(5) *By the way, the authors claim that their equation of motion is the first relativistic one; this is not correct, as Bohm proposed a relativistic equation of motion on the basis of a Dirac wave function in Prog. Theor. Phys. 9: 273 (1953), also described in Chap. 12 of [6].*

Response:

In light of our response to referee 1 and some of the comments made above, we maintain the accuracy of our claim, that

A consistent theory of Bohmian particle trajectories based on *observed* velocity fields in relativistic regimes has not been developed (emphasis added).

This is consistent with our primary motivation in constructing an operationally-based relativistic formulation of Bohmian mechanics, with the possible future experimental verification of such trajectories in mind. Likewise, we state

In this article we propose a velocity equation describing the Bohmian trajectories of relativistic particles, specifically photons possessing a relativistic energy dispersion. Our new velocity equation can be derived operationally through weak measurements of the particle momentum and energy, which we respectively identify with the Klein-Gordon conserved probability current and density.

which we also maintain is an accurate representation of what our paper claims to achieve (bearing in mind that we are using a scalar-valued wavefunction to model photons, as discussed above). That is, the derived velocity field, obtained from the operational weak measurement formalism, is the first of its kind.

We have added additional sentences discussing the historical attempts that have been made to extend Bohmian mechanics to relativistic regimes in the introduction (page 2, left-hand column). Likewise we have made the novel aspects of our paper clearer within these paragraphs.

• Comment:

I do not see anything substantial added to the literature by the manuscript. I recommend rejection.

Response:

We appreciate the time taken for the referee to review our manuscript. We also hope that our responses persuade the referee that our approach and results are indeed novel and that it will be a paper of high impact within the scientific community. To summarise:

1. Comments (1), (3) and (4) concern the perceived lack of generality of our paper. In response, we have not claimed to derive a new physical law but have reformulated Wiseman's nonrelativistic theory to account for particles with relativistic energies. We have justified our physical assumptions, showing that they are consistent with our measurement approach, the domain of applicability of our theory, and foreseeable applications to experiment.
2. Comments (2) and (5) raise concerns over the advance of our findings compared with prior works. In line with our response to referee 1, prior works are disconnected from an operational formalism which ground the results in terms of physical measurements (which Wiseman's approach does). Many pathological issues which these papers face (including those cited by referee 1) can be explained when basing these trajectories in weak measurements. We have duly acknowledged these works which have been important in the progression of the field.

Because of these reasons, we respectfully ask the referee to reconsider our paper and its place in *Nature Communications*.

Referee 3

- Comment:

The central question to this work is the introduction of a relativistic expression for the Bohmian guidance condition, from which trajectories describing the evolution of the quantum system can be synthesized after integration. More specifically, what makes appealing the equation determined is the fact that it would allow the inference of Bohmian trajectories for photons in the relativistic regime by performing weak measurements of both momentum and energy, as it has already been done by Steinberg's group in Toronto in the nonrelativistic regime. Furthermore, a simple model to extend this equation to the domain of general relativity is also suggested by considering a modified Alcubierre metric, although this part is only slightly elaborated.

The idea is interesting, because, to the best of my knowledge, there have not been proposals to tackle the issue of experimentally determining (even though only at an inferential level, as in the case of the experiments mentioned before) Bohmian-type quantum trajectories in the relativistic domain making use of weak measurements. The chosen system to proceed, as I understand, would be photons, which increases the interest of the proposal, because the weak-measurement techniques are quite developed for photons.

Response:

We thank the referee for acknowledging the potential interest and impact of our work and in both the theoretical and experimental communities.

- Comment:

However, in my opinion, there are some questions or aspects that need first to be properly clarified and/or conveniently revised, which might contribute to improve the context, understanding and potential reach and interest of the work:

Response:

We address these concerns below.

- Comment:

At the conceptual level, although Bohmian mechanics is widely regarded as a deterministic, nonlocal theory, as it is also taken for granted here, note that it is nothing but another reformulation of quantum mechanics. As Bohm himself acknowledges, what he did was to rationalize what a quarter of a century earlier de Broglie had

only postulated, namely that quantum systems can be understood as point-like particles (like any classical system) acted by a wave field (making here a capital difference with classical systems), apart from eventual external fields. There is still an on-going debate on whether both approaches are the same or, on the contrary, there are some divergences, but this is a minor issue to the discussion here (actually, totally irrelevant). This idea came up after Bohm's attempts to disprove the conclusion of von Neumann's theorem on the impossibility of a hidden-variable-based explanation of quantum phenomena, as the authors might know well.

So far, so good, except for the fact that, alternatively to de Broglie, the same year that Schrödinger published his equation, Madelung sent to publication a manuscript with a reformulation of the former's equation in a hydrodynamic form [Z. Phys. 40, 322 (1927)]. By means of a polar ansatz identical to the one later on considered by Bohm, but directly associating it with a system energy (to some extent, in the Hamiltonian way), he concludes how the Schrödinger equation should be reinterpreted in hydrodynamic terms, emphasizing towards the end, for instance, that the so-called quantum jumps should be understood in a continuous rather than discrete manner. Some other conclusions linking the probability density with ensembles of identical realizations are also worth noting, though.

Accordingly, it seems there is nothing new in Bohmian mechanics as a theory, as widely claimed in the literature (except by Bohm himself, who indeed disliked the term "Bohmian mechanics"), but just another formulation of the standard quantum mechanics, although less used because of unjustified prejudices, particularly, the fact that it incorporates the notion of well-defined trajectory. But there is nothing wrong about this, for in the same way that hydrodynamics includes streamlines that serve to monitor the flow of a fluid, the so-called Bohmian trajectories are streamlines that serve to monitor the flow of the probability density.

In sum, although the manuscript tries to, somehow, put the emphasis on the old idea of hidden variables and alternative quantum theories, this is nothing but introducing more of unnecessary noise in the literature on the issue.

Response:

We appreciate the referee's deep knowledge of the development of this field. We agree that our original introduction may have overemphasised aspects of the historical development which our paper is less concerned with. We have restructured the introduction so that it more clearly and concisely communicates our desired emphasis on the novel operational nature of our investigation. As the referee understands, Bohmian mechanics is one of many interpretations of quantum mechanics. However it is rather unique in admitting this operational measurement formalism for inferring the deterministic trajectories of particles which exist in the interpretation.

• Comment:

In connection to the previous point it should also be stressed that, contrary to the authors' claim, quantum (Bohmian) trajectories have not been observed experimentally, but they have only been experimentally inferred. This is a totally different question, which has induced to misconceptions in the literature and related debates at conferences regarding Steinberg's experiments. In such experiments, with the aid of both weak and strong (von Neumann) measurements the transverse flux and the probability density are respectively determined. From those experimental data, "taking for granted" a Bohmian-based guidance equation, the corresponding trajectories were inferred. However, such trajectories do not represent the paths followed by the photons themselves, but only streamlines along which the electromagnetic energy flows. This is something somehow acknowledged in the title of Steinberg's Science paper, Ref. 18, by denoting them as "average trajectories", which takes us back again to Madelung's viewpoint. Note that there is no empirical evidence that a single photon can be uniquely associated with a given trajectory, because weak measurements determine, in the end, average values, even though in a particular manner.

Response:

As discussed, the interpretation of the Steinberg experiment is not a direct issue we address in our paper. Nevertheless, it is clear that the significant interest that the paper has attracted has been the attachment of a real measurement model to the inference of these particle trajectories. The crux of our work is to show how to apply this operational approach in the construction of *relativistic* particle trajectories.

Throughout the paper, we have changed the language of 'observed' and 'determined' to 'inferred trajectories'.

• Comment:

Furthermore, it is worth mentioning that, contrary to a general believe, the possibility to measure other quantities than the usual expectation values, based on the notion of von Neumann's strong measurement, is already incipient in Hirschfelder's equations of change, suggested by the late 1970s [J. Chem. Phys. 68, 5151 (1978)]. These equations are intended to measure, for instance, the flow of energy in the system from one part to another one, which is actually what Bohmian trajectories provide us with in the case of photons (electromagnetic fields).

Response:

Thank you for pointing this out. In our view Hirschfelder's work is quite distant from connecting a weak measurement model with the construction of a relativistic Bohmian velocity field. We have included it in the reference list of our paper.

- Comment:

Regarding the approach considered here, what calls my attention is the fact that the authors wish to determine photon trajectories (understand this in line with the above comments), but they make use of the Klein-Gordon equation, which is only applicable to scalar (spinless) particles. It is clear that the framework it provides helps to “capture the essential Lorentz relativistic properties of photons”, as it is acknowledged by the authors. However, photons require a more complete treatment, such as the Dirac equation (a QED extension of Maxwell’s equations), at least like the one provided about twenty years earlier by Ghose and coworkers (Ref. 23), based on Dirac’s equation. Other important works developed by Dewdney and Horton or Nikolic on relativistic extensions of Bohmian mechanics, published along the 2000s, seem also to have been disregarded, although they constitute an appropriate theoretical ground.

Response:

This is related to referee 2’s comments concerning the modelling of photons (page 5 and 6 of this letter). We agree that a more developed theory could incorporate an extension to spin-1/2 or spin-1 particles. However as we stated above and in our paper, scalar quantum mechanics/field theory is the most basic building block for modelling relativistic particles which to a great extent captures the main features of a vector theory. As we also discuss in our response to referee 2, when $k \gg k_z$, adding vector components to describe polarisation does not significantly change the underlying physics in regards to the application of relativistic covariance.

The changes made to our paper in view of this are stated on page 6 of this letter.

- Comment:

Some of those works, though, directly attack the Klein-Gordon equation and provide a proper guidance equation [Found. Phys. 32, 463 (2002); J. Phys. A 33, 7337 (2000); Found. Phys. Lett. 18, 549 (2005)]. Same happens with the more recent work by Capistrano, cited in Ref. 24 (please, remove Jalalzadeh, since he finally did not coauthorized the published version of the paper). However, it seems none of them have been seriously considered as a starting point (even though with the aforementioned limitations for photons).

Response:

We emphasise that the starting point in deriving the Bohmian trajectories is not the Klein-Gordon field/equation but rather our weak measurement model (we likewise address this in our response to referees 1 and 2, for example page 3 of this letter). The appearance of the conserved current/density gives rise to natural relativistic

properties of the velocity field. The foundational motivation for our paper is a reformulation of Wiseman's weak measurement framework which includes relativistic effects.

Regarding the mentioned references, we thank you for pointing these out. We have removed Jalalzadeh from Ref. 24 and included additional references where appropriate. In response:

- Found. Phys. 32, 463 (2002): this paper focuses on the pathologies of the Klein-Gordon theory, in particular the issues arising from the non positive-definite current density. Our weak measurement approach allows for a unique operational interpretation of such densities (discussed further on page 8 and 9 of this letter). We have included this as a reference on page 2.
- J. Phys. A. 33, 7337 (2000): this paper takes the common approach by writing the wavefunction in polar form and obtaining a corresponding velocity equation. The key difference with our paper is again, that we tackle the problem from an operational angle, obtaining a measured velocity field. We have included this in a reference on page 2.
- Found. Phys. Lett. 18, 549 (2005): this paper was cited by referee 1. Again, the treatment is the more 'standard' Bohmian approach which is disconnected from any measurement model. We have likewise included this reference on page 2.

• Comment:

The above point poses another important issue: although Bohm introduces the so-called guidance equation as a postulate, this is not an essential requirement to introduce (Bohmian) trajectories, for such an expression directly and naturally arises from the fact that a transport equation rules the diffusion of the probability density in configuration space, namely the continuity equation. Accordingly, the quantum flux is directly related with the probability density by means of a velocity field. This basic relation also appears in other wave theories, which enables the extension of the same ideas to such theories (or, in other words, to extrapolate notions from those other theories to the quantum realm, essentially what Madelung did). This is an important issue, because it directly implies that, whenever we are able to find a transport equation in quantum mechanics, we can describe the phenomenon in a hydrodynamic fashion. Accordingly, there must be a unique relationship between a density function and the corresponding flux.

Although it is mentioned that the flux and probability density here satisfy a continuity equation, namely Eq. (12), it is not clear where does Eq. (6) exactly come from?

If one goes to Sec. IIIA, “Methods”, it seems that Eq. (32) [the same as Eq. (6)] does not arise as a direct consequence of a transport process, but it is “proposed”. This point should be properly clarified. Otherwise, it constitutes an important break from the ideas underlying Bohmian mechanics (some of them have already been slightly sketched in the comments above), entering the realm of speculation.

Response:

Thank you for the insightful comments.

Eq. (5) of the manuscript gives the velocity constructed from the ratio of two components of the energy-momentum 4-(2-)vector. Given the worldline of a particle, this quantity is expressed as the product of the particle rest mass and the tangent vector along the curve given the curve is parameterised by the proper-time. The ratio of the two quantities only depends on the actual parameterisation in so much as the curve depends on the coordinates chosen. Therefore any parameterisation affinely related to the proper-time will give the same value for $V(x, t)$ as shown in Eq. (5).

Wiseman gives a nonrelativistic expression for a tangent vector related to an observable through weak-values (NJP (9), 165 (2007), Eq. (6)). The relativistic energy-momentum 4-(2-)vector can be similarly expressed under the conditions of affinely related proper-time given above.

Using this weak-valued expression gives a weak-valued 4-(2-)vector expression for the components in Eq. (5). This naturally leads to our Eq. (6), where one uses the standard definition of the weak-value (Eq. (2) of the resubmitted manuscript).

Like Wiseman, one must still interpret the weak-values as a measurement of the velocity field. However, we naturally have an expression for the guiding velocity that gives integral curves which exhibit Lorentz covariance as the velocity expression exhibits the correct transformation rules for velocities. This is shown in great detail through the parts of our paper subsequent to our introduction of this relativistic weak-valued velocity equation.

- Comment:

On another note, it is found that Eq. (6) is given in terms of momentum divided by energy. This sounds odd to me, particularly in the case of light, where one would expect energy (or frequency) over momentum (or wavevector), which has the dimensions of a velocity (actually, in vacuum, we should obtain something proportional to c) both in quantum mechanics and also in special relativity.

Response:

Eq. (6) is our proposed weak-value defined velocity equation. In our exposition we have used units where $c = 1$. This is a convenient choice to remove any clutter due to conversion constants in equations but comes with the drawback that some

dimensional analysis can be misleading. Equation (6), like Eq. (5), comes from taking components of the energy-momentum four-vector. Reinserting factors of c gives the dimensionless velocity $V(x, t)/c = p(x, t)c/E$ which has the dimensions that the referee is expecting.

- Comment:

In nonrelativistic Bohmian mechanics the r.h.s. of the guidance equation is directly related with the real part of the local evaluation (in space) of the usual momentum operator. Here, I am afraid that I might be missing something, so I would acknowledge a clarification.

Response:

The crux of Wiseman's original argument in NJP (9), 165 (2007) was that one can interpret the local evaluation of the momentum operator via a weak measurement. That is, Eq. (6) and then (7) of Wiseman's paper are the nonrelativistic versions of our proposal.

- Comment:

Furthermore, should not the velocity be defined, in general, in terms of a proper time rather than the usual space and time coordinates? It seems that here time has also been used as an evolution parameter instead of a variable itself, such as the spatial coordinate, thus making little distinction with nonrelativistic quantum mechanics.

Response:

It is possible use a local coordinate system following the path of a single particle. However as mentioned, we are interested in an operational measurement approach which does not need to be defined in terms of the proper time, τ . Instead, we construct the coordinate velocity field $V(x, t)$ from measurements of energy and momentum in a particular reference frame which avoids the need for an additional evolution parameter. The plotted trajectories are simply obtained from the integral curves of the velocity field.

We did not define the proper time τ in our original submission – we have done that now on page 3. Likewise we have emphasised throughout that the obtained velocity field is a coordinate velocity, for example page 3, top of right-hand column, page 8, bottom of left-hand column.

- Comment:

Although in the relativistic domain, the analysis of head-on collisions is interestingly similar to the one published earlier on by Sand and Miret-Artés about wave-packet

head-on collisions [J. Phys. A 41, 435303 (2008)], including the α parameter specifying the contribution from each side, which is not included here in the reference list. Independently of this, it is interesting to note that the differences with that work seem to be only the origin of time, for the trajectories (and the behavior of the probability densities) look pretty similar to the results expected in a nonrelativistic scenario, like the one considered by Sand and Miret-Artés. Actually, in the latter case, a simple translation in time would render similar figures. A direct comparison cannot be set, because the density is omitted in the referred work, but the behavior of the trajectories is similar.

Response:

Thank you for pointing out this paper to us – we have included this in the updated version on page 1.

The mentioned paper studies the trajectories in the nonrelativistic (massive, $k_z \gg k$) limit regime, while our focus is on the relativistic ($E(k) = k^2 + k_z^2$ and $E(k) \propto |k|$) regime. A closer comparison would be between the trajectories plotted in the Sand/Miret-Artés paper and those shown in our section on the paraxial limit, which uses a nonrelativistic form of the energy to obtain the particle trajectories. Finally, care must be taken when comparing their nonrelativistic trajectory plots with ours in the relativistic limit, because the apparent qualitative similarity appears because the units of time are not specified in their paper.

In addition to the stated changes, we have included a sentence on page 5 explaining the relativistic normalisation of the Klein-Gordon probability density, and fixed some minor typos (all highlighted in blue in the revised manuscript).

We would like to thank all referees again for taking the time to review our manuscript. We believe the manuscript has gained further clarity and will be received with great interest by a wide audience if published. We also believe we have adequately addressed all of the concerns that the referees have raised, and thus believe it is worthy of publication in *Nature Communications*.

We are looking forward to your response.

With kind regards,
Joshua Foo
(On behalf of all authors)

joshua.foo@uq.edu.au
+61 432 536 832

ARC Centre for Quantum Computation and Communication Technology (CQC2T)
University of Queensland,
Australia

REVIEWER COMMENTS

Reviewer #1 (Remarks to the Author):

The authors have satisfactorily responded to my objections, so I recommend publication.

Reviewer #2 (Remarks to the Author):

In the revision, the authors have given a key role to weak measurements, which addresses some of my criticisms. However, I still have several major reservations:

1) The trajectories for Klein-Gordon wave functions considered here have long been known and are physically unconvincing because they can have stretches that can be spacelike or even past-timelike. Of the last fact, the authors seem unaware.

2) Although the authors cite prior works on relativistic Bohmian trajectories on p 2 of their manuscript, in the abstract they continue to pretend that they are considering relativistic Bohmian trajectories for the first time, whereas in fact Bohm himself presented trajectories for the Dirac equation in 1953. I find this unacceptable for a scientific publication.

3) That Bohmian trajectories can be constructed from weak measurements has been pointed out by Wiseman in 2007 for non-relativistic Bohmian mechanics. That this can be done as well for Klein-Gordon wave functions is new (as far as I know) but not unexpected.

4) The modified Alcubierre-type metric introduced in Section I.D is not what observers would regard as “the” space-time metric; it must be thought of as a “made-up” metric. Introducing this metric does not make the unconvincing features of the trajectories go away.

A few more detailed comments:

1.1) The authors, writing in their reply to the referees (p 2) that “backwards-in-time trajectories arising from boosts can be understood as an artifact of the choice of reference frame,” seem unaware that the j^μ vector field (11) to which the trajectories are tangent can also be past-timelike, and thus point to the past in every frame; as a consequence, some trajectories for some wave functions ψ will have past-timelike pieces (even if ψ contains no contributions of negative

energy). Correspondingly, the statement on p 8 of the reply that “it is always possible to perform a global coordinate transformation where the velocity field is forward directing” is mistaken.

1.2) On p 7 of their reply, the authors say they couldn't find their equations (10) and (11) in the Bohm-Hiley book. I see that in the 2009 e-book edition, pages are numbered differently. As I said in my previous report, (10) and (11) are equivalent to (11.2) and (11.3) in the book, which are in Section 11.2, now presumably on p 195 or 196. I was not referring to equations (12.1)-(12.3).

1.3) On p 2 of the manuscript, the authors say that the “missing link between [prior works] and a consistent interpretation of relativistic Bohmian mechanics is the notion of operationalism[.]” Likewise in their reply to the referees, they emphasize that the trajectories should be regarded “through the lens” of their construction from weak measurements. This presumably means that the trajectories should not be taken seriously as the actual world lines of real particles. But that undercuts the interest in the question. If the particles are fictitious and the trajectories not serious, then why bother with them?

2.1) In the abstract, the authors write: “Historically, the study of Bohmian trajectories has been restricted to nonrelativistic regimes due to the widely held belief that the theory is incompatible with special relativity. Here we derive expressions for the relativistic velocity field and spacetime trajectories . . .” This could hardly be understood differently than as saying that the present paper of Foo et al. is the first to introduce relativistic Bohmian trajectories. This is absolutely inappropriate in view of Bohm's 1953 relativistic trajectories for Dirac particles (by the way, not cited in the manuscript) and the fact that the very trajectories considered here were discussed in the 1993 book of Bohm and Hiley (and presumably also already way earlier than that).

3.1) The authors say in their reply that, because it is based on weak measurements, their approach is “fundamentally an operational one.” Well, this wording is rather inflated because the approach is really guessing trajectories from empirically accessible data. It is no more operational than starting from a probability current 4-vector field j^μ , which after all can be measured in an ensemble by measuring the probability density $\rho = j^0$ in every Lorentz frame (on every spacelike hyperplane) and computing j^μ from that, and defining the velocity v^i as j^i/j^0 (as suggested by Bell in 1965).

4.1) What if several photons with different wave functions are present? Then we have many metrics to choose from. Specifically, if two space-time points x, y are spacelike separated relative to the usual Minkowski metric but not relative to the modified metric, then either it is still impossible to send signals from x to y or long-standing principles of physics would be overthrown in a spectacular way—which would need solid evidence. What my example illustrates is that the ordinary Minkowski metric, and not the modified metric, is the one relevant to defining the relation “spacelike.”

To sum up, while some of the material may be worth publishing as further properties or alternative derivations of the questionable current formula (11), the manuscript in its present form is not acceptable.

Reviewer #3 (Remarks to the Author):

I am satisfied with the reply provided by the authors to the questions and comments posed in my report as well as with the new version of the manuscript, which, overall, is clearer and more consistent than the previous one. In my opinion, this version is now suitable for publication.

There is, though, a very minor issue. As far as I have noticed, refs. 41 and 42 do not appear anywhere in the main text. Please, check this.

Response to Reviewers

Referee 1

• Comment:

The authors have satisfactorily responded to my objections, so I recommend publication.

Response:

We thank the referee for recommending publication.

Referee 2

• Comment:

In the revision, the authors have given a key role to weak measurements, which addresses some of my criticisms. However, I still have several major reservations:

Response:

We thank the referee for acknowledging the clarifications and improvements made in our first response, and are glad that it addressed some of the criticisms raised.

• Comment:

1) The trajectories for Klein-Gordon wave functions considered here have long been known and are physically unconvincing because they can have stretches that can be spacelike or even past-timelike. Of the last fact, the authors seem unaware.

1.1) The authors, writing in their reply to the referees (p 2) that "backwards-in-time trajectories arising from boosts can be understood as an artifact of the choice of reference frame," seem unaware that the j_μ vectorfield (11) to which the trajectories are tangent can also be past-timelike, and thus point to the past in every frame; as a consequence, some trajectories for some wave functions will have past-timelike

pieces (even if ψ contains no contributions of negative energy). Correspondingly, the statement on p 8 of the reply that "it is always possible to perform a global coordinate transformation where the velocity field is forward directing" is mistaken.

Response:

We are aware of this issue, and the referee raised a very similar concern in the first round of review. The quotations cited in 1.1) are true under the ‘optical approximation’ which we employ throughout our paper.

Firstly, it appears that the referee doesn’t appreciate our use of the approximation, which is perhaps based in a misunderstanding of what it means physically. The utilisation of the approximation is simply a restatement of the fact that our proposal is a relativistic *single-particle* theory. Leaving the domain of the optical approximation (i.e. $O(k_0) \sim O(\sigma)$) describes the physical regime directly associated with subcycle particle production (i.e. multiparticle effects) in quantum electrodynamics. We have explained this in detail previously – see Pages 8 and 9 of our revised manuscript, or Pages 8 and 9 of our previous reply. We have mentioned that extending our theory to include multiple particle effects such as particle production is beyond the scope of the current investigation.

Secondly, once one applies our approximation, the statement

‘it is always possible to perform a global coordinate transformation where the velocity field is forward directing’

and likewise

‘backwards-in-time trajectories arising from boosts can be understood as an artifact of the choice of reference frame’

can be straightforwardly proven to be true. Conversely, the referee’s statement

‘... the j_μ vector field (11) to which the trajectories are tangent can also be past-timelike, and thus point to the past in every frame...’

is provably false under the optical approximation. We outline this proof below, which we have incorporated in full detail in the Methods section of our resubmitted manuscript.

1. For the velocity field to possess past-timelike tangent vectors, the following conditions must be simultaneously satisfied:

$$\dot{f}^2(x, t) - \rho^2(x, t) < 0 \quad \text{and} \quad \rho(x, t) < 0. \quad (1)$$

If the first condition is satisfied, the tangent vector at (x, t) is timelike (i.e. the magnitude of the metric length is negative) while if the second condition is satisfied, the tangent vector points to the past. Therefore the sufficient condition for an everywhere forward-directed velocity field is that $\rho(x, t) \geq 0$ everywhere.

The referee is concerned that there might be some choices of $\psi(x, t)$ which yield past-timelike tangent vectors, meaning that no global coordinate transformation is able to ensure that the velocity field is forward-directing everywhere. To prove that such a global coordinate transformation *does* exist, it is sufficient to show that there always exists a reference frame in which $\rho(x, t) \geq 0$ everywhere, and therefore that the velocity field is forward-directing everywhere in that reference frame. This would concurrently prove that in the optical approximation, the velocity field does not yield past-timelike tangent vectors in any frame.

2. We have already shown that when the left- and right-moving parts of $\psi(x, t)$ have both equal k 's and σ 's, the condition

$$\rho(x, t) \geq 0 \quad \text{for all } (x, t) \quad (2)$$

is true under the optical approximation. This is explained in the paragraph following Eq. (22) of our manuscript.

3. The only other scenario is when the k 's and σ 's of the left- and right-moving parts of $\psi(x, t)$ are unequal. Such a scenario mimics one in which the left- and right-moving k 's and σ 's (denoted (σ_L, k_{OL}) and (σ_R, k_{OR}) respectively), are Doppler shifted (i.e. boosted). In such a scenario, there may arise points where $\rho(x, t) < 0$ (as discussed on Page 8 and 9 of our manuscript, negative densities arising from boosts are a natural consequence of the superluminal trajectories arising in Bohmian mechanics).

In our revised Methods section, we show that there always exists a boost which makes

$$k'_{OR} = k'_{OL} \quad \text{with} \quad \sigma'_R \neq \sigma'_L \quad (3)$$

or

$$\sigma'_R = \sigma'_L \quad \text{with} \quad k'_{OR} \neq k'_{OL} \quad (4)$$

(the primes denote that we have boosted to a new reference frame). We likewise show that if the optical approximation holds in a particular reference

frame, then it holds in any reference frame. Applying the optical approximation to $\rho'(x', t')$ in the boosted reference frame where $k'_{0R} = k_{0L}$, we prove that

$$\rho'(x', t) \geq 0 \quad \text{for all } (x', t) \quad (5)$$

This shows that in the optical approximation, our prior assertion is indeed true:

‘it is always possible to perform a global coordinate transformation where [the density is positive, there are no past timelike pieces and hence], the velocity field is forward directing’.

This addresses the concern raised by the referee regarding the technical validity of our proposal.

As mentioned, we have added this discussion in full detail in the Methods section of our paper.

• Comment:

1.2) On p 7 of their reply, the authors say they couldn’t find their equations (10) and (11) in the Bohm-Hiley book. I see that in the 2009 e-book edition, pages are numbered differently. As I said in my previous report, (10) and (11) are equivalent to (11.2) and (11.3) in the book, which are in Section 11.2, now presumably on p 195 or 196. I was not referring to equations (12.1)-(12.3).

Response:

Thank you for clarifying this. As explained on Page 7 of our first round reply letter, the fact that an equation of the form $V(x, t) = j_{KG}(x, t)/\rho_{KG}(x, t)$ emerges from our weak-value prescription shows that it is indeed consistent as a relativistic theory.

We note that after introducing Eq. (11.2) and (11.3), Bohm states that ‘we cannot make a consistent particle interpretation of the kind given for the Klein-Gordon equation [i.e. 11.2 and 11.3]’ (emphasis added). Our measurement-based proposal gives this interpretation – it is a consistent description in the single-particle regime as described by the optical approximation.

We have added two references to Bohm’s book in the second paragraph on Page 2.

• Comment:

1.3) On p 2 of the manuscript, the authors say that the “missing link between [prior works] and a consistent interpretation of relativistic Bohmian mechanics is the notion of operationalism[.]” Likewise in their reply to the referees, they emphasize

that the trajectories should be regarded "through the lens" of their construction from weak measurements. This presumably means that the trajectories should not be taken seriously as the actual world lines of real particles. But that undercuts the interest in the question. If the particles are fictitious and the trajectories not serious, then why bother with them?

Response:

We disagree with the referee's presumption that 'the trajectories should not be taken seriously as the actual worldlines of real particles'. We elaborate on our use of the term 'operational' on Page 6 of this response, but in short: the weak-value measurement construction has the advantage of being explicit in how one would translate the meaning of the trajectories onto the outcomes of a measurement apparatus. All of the coordinates describing the measurement apparatus and the trajectories it reconstructs are expressed in terms of spacetime variables. Hence one may make the link to a process being carried out intrinsically within spacetime. We do not need to invoke trajectories within phase-space or any other constructions which would make the interpretation of such trajectories problematic.

• Comment:

2) Although the authors cite prior works on relativistic Bohmian trajectories on p 2 of their manuscript, in the abstract they continue to pretend that they are considering relativistic Bohmian trajectories for the first time, whereas in fact Bohm himself presented trajectories for the Dirac equation in 1953. Ifind this unacceptable for a scientific publication.

2.1) In the abstract, the authors write: "Historically, the study of Bohmian trajectories has been restricted to nonrelativistic regimes due to the widely held belief that the theory is incompatible with special relativity. Here we derive expressions for the relativistic velocity field and spacetime trajectories. . . ." This could hardly be understood differently than as saying that the present paper of Foo et al. is the first to introduce relativistic Bohmian trajectories. This is absolutely inappropriate in view of Bohm's 1953 relativistic trajectories for Dirac particles (by the way, not cited in the manuscript) and the fact that the very trajectories considered here were discussed in the 1993 book of Bohm and Hiley (and presumably also already way earlier than that).

Response:

We acknowledge that the language in the abstract did not capture the intent of the revisions which we made to the introduction in the first round, and so it appears to claim something more general. Our statements in the abstract were supposed to be understood in the context of weak-measurement trajectories.

We have now changed the abstract to remove any misinterpretation of this kind. We have also cited Bohm's 1953 paper on Page 1 and 2 of the revised manuscript. We have added two sentences on Page 2 mentioning attempts at extending Bohmian mechanics to relativistic Dirac particles (including Bohm's paper), and the issues therein.

• Comment:

3) That Bohmian trajectories can be constructed from weak measurements has been pointed out by Wiseman in 2007 for non-relativistic Bohmian mechanics. That this can be done as well for Klein-Gordon wave functions is new (as far as I know) but not unexpected.

Response:

This comment suggests that what we describe indeed must be true and correct, and so thank the referee for this affirmation. Although the referee says that this is not unexpected, we note that Wiseman himself did not expect this to be possible, as pointed out in a footnote of his original paper (see Wiseman NJP 9 165, 2007, pp. 10).

• Comment:

3.1) The authors say in their reply that, because it is based on weak measurements, their approach is "fundamentally an operational one." Well, this wording is rather inflated because the approach is really guessing trajectories from empirically accessible data. It is no more operational than starting from a probability current 4-vector field \mathbf{j}_μ , which after all can be measured in an ensemble by measuring the probability density $\rho = \mathbf{j}^0$ in every Lorentz frame (on every spacelike hyperplane) and computing \mathbf{j}_μ from that, and defining the velocity \mathbf{v}^i as $\mathbf{j}^i/\mathbf{j}^0$ (as suggested by Bell in 1965).

Response:

By 'operational', we mean a precise, step-by-step set of instructions using fundamental concepts of measurement and time-evolution. In this way, our theory can be directly applied to the construction and analysis of an experiment (under ideal conditions). The process of weakly measuring observables is itself operationally defined within a measurement model. With the descriptions in our manuscript, all steps for an experiment to reconstruct the velocity field and the corresponding trajectories are spelled out. If one could directly measure the probability current then the referee's comment would be apt. As the referee suggests, one could reconstruct a current from ensemble averages, however this is not an operational definition, as no measurement process nor a method of processing the data is described. In contrast, our paper gives all of this information.

We have added a footnote on Page 1 explaining our definition of the term ‘operational’.

• Comment:

4) The modified Alcubierre-type metric introduced in Section I.D is not what observers would regard as "the" space-time metric; it must be thought of as a "made-up" metric. Introducing this metric does not make the unconvincing features of the trajectories go away.

Response:

The metric introduced in this section was intended to describe how a natural generalisation of the concept of a guiding force to that of a guiding metric can be made. It shows that even in the presence of our unusual trajectories, such a metric exists. This observation could be useful for later work when considering couplings (e.g. conformal coupling) in a field theory should such an extension to our approach be considered. Our consistent general relativistic way of describing the curvature of the trajectories can feature backwards-in-time tangent vectors in some reference frames.

We have added a sentence on Page 7 reemphasising this point.

• Comment:

4.1) What if several photons with different wave functions are present? Then we have many metrics to choose from. Specifically, if two space-time points x, y are spacelike separated relative to the usual Minkowski metric but not relative to the modified metric, then either it is still impossible to send signals from x to y or long-standing principles of physics would be overthrown in a spectacular way—which would need solid evidence. What my example illustrates is that the ordinary Minkowski metric, and not the modified metric, is the one relevant to defining the relation "spacelike."

Response:

That our proposal is only a single particle theory is a clear limitation of what we present in the manuscript. The logic the referee develops is therefore outside the scope of what we present here, though these would be natural questions to be asked going forward. Moreover, we do not see any fundamental reasons why a generalisation to multiple particles is not possible.

We have added a sentence on Page 9 suggesting this as a question for future investigations.

• Comment:

To sum up, while some of the material may be worth publishing as further properties or alternative derivations of the questionable current formula (11), the manuscript in its present form is not acceptable.

Response:

We respectfully disagree that our results are merely an alternative derivation of a questionable formula. Wiseman's 2007 paper was the first to ground nonrelativistic Bohmian particle trajectories in the context of an operational measurement framework. This in turn motivated two paradigmatic experiments by Kocsis et. al. (Science 332(6034), 2011) and Mahler et. al. (Science Advances 2(2), 2016) which observed such trajectories in an experiment for the first time. As stated in our manuscript and previous reply, our paper reformulates Wiseman's weak-value prescription to be consistent with special relativity, overcoming the problems and/or inconsistencies encountered in prior attempts (such as those mentioned by the referee).

Referee 3

• Comment:

I am satisfied with the reply provided by the authors to the questions and comments posed in my report as well as with the new version of the manuscript, which, overall, is clearer and more consistent than the previous one. In my opinion, this version is now suitable for publication.

There is, though, a very minor issue. As far as I have noticed, refs. 41 and 42 do not appear anywhere in the main text. Please, check this.

Response:

We thank the referee for recommending publication. We have addressed the issue by including citations of [41] and [42] on Page 2 and 1 respectively of the revised manuscript.

We thank all referees and Dr. Bentivegna for again taking the time to review our manuscript. We believe we have adequately addressed all of the concerns that the referee has raised, and maintain that our manuscript is worthy of publication in *Nature Communications*. We list below all the revisions made to our manuscript (also highlighted blue in the resubmission).

• **Abstract:** changed

‘the study of Bohmian trajectories has been restricted to nonrelativistic regimes...’

to:

‘the study of Bohmian trajectories has **mainly** been restricted to nonrelativistic regimes...’

- **Abstract:** we have added the following modification to reflect the intended emphasis of our paper.

‘Here, we present a new approach for constructing the relativistic Bohmian-type velocity field of single particles. The advantage of our proposal is that it is operational in nature, grounded in weak measurements of the particle’s momentum and energy. We apply our weak measurement formalism to obtain the relativistic spacetime trajectories of photons in a Michelson-Sagnac interferometer.’

- **Page 1:** added a reference to Bohm’s 1953 paper raised by referee #2.
- **Page 1:** added the following footnote clarifying the definition of ‘operational’.

‘By operational, we mean a precise, step-by-step set of instructions using only fundamental concepts such as measurement and time-evolution.’

- **Page 2:** included the intext citations to [41] and [42] mentioned by referee #3.
- **Page 2:** included an additional reference to the Bohm-Hiley book ([6]) as an example of a prior study using the Klein-Gordon theory which carries interpretive issues.
- **Page 2:** added the following sentences mentioning the persistent inherent difficulties of extending Bohmian mechanics to relativistic Dirac particles.

‘Other works have used to use the relativistic Dirac equation as the basis of constructing Bohmian trajectories for spin-1/2 particles, however these studies are not without their own issues. For example, Nikolic’s formulation requires the postulation of additional hidden-variables, whose physical interpretation is not clear.’

- **Page 2:** changed ‘obtained’ to ‘**defined**’ to emphasise the operational starting point of our proposal.
- **Page 2:** added the following comment clarifying the role of the optical approximation.

‘(hence the optical approximation implies a single-particle description)’

- **Page 7:** added the following sentence explaining the meaning of the ‘photon metric’.

‘That such a metric exists in the presence of our unusual trajectories could be useful for later work when considering couplings (e.g. conformal coupling) in a field theory, should such an extension to our current single-particle theory be considered.’

- **Page 9:** we have added the following sentence on Page 9.

‘Likewise, it would be worthwhile understanding how our interpretation of the ‘guiding metric’ might be generalised for multiple particles.’

- **Page 11 and 12:** added the following sentence to point out the extension of our ‘guiding metric’ proposal to multiple particles, suggested by referee #2.

‘Likewise, it would be worthwhile understanding how our interpretation of the ‘guiding metric’ might be generalised for multiple particles.’

- **Page 11 and 12:** we have added the proof outlined on Page 2-4 of this letter in the Methods section (Page 11 and 12).

We are looking forward to your response.

With kind regards,
Joshua Foo
(On behalf of all authors)

joshua.foo@uq.edu.au

+61 432 536 832

ARC Centre for Quantum Computation and Communication Technology (CQC2T)
University of Queensland,
Australia

REVIEWERS' COMMENTS

Reviewer #1 (Remarks to the Author):

I looked at the changes in the manuscript as responses to objections by another (not me) referee. In my opinion the responses are satisfying and the paper can be published.

Reviewer #2 (Remarks to the Author):

With the present revision, I am still not enthusiastic about the paper. But my biggest concern, that prior works on relativistic Bohm trajectories were not properly credited, has been taken care of. The authors may benefit (for this paper or future ones) from a few more comments:

A) The authors seem to think that world lines turning around in space-time (e.g., U-shaped world lines) somehow represent particle creation (respectively, annihilation) events. But that would be a wrong treatment because a particle pair gets guided by a two-particle wave function, not by the one-particle wave function we are considering, and its probability density is a function $\rho(x_1, x_2)$ of two locations x_1 and x_2 .

B) The authors declare that they take the trajectories seriously as the world lines of real particles. Good. But in view of the emphasis on operational considerations, half of all readers will think that the trajectories are just a mathematical fiction and not real.

C) The authors dismiss the U-shape problem saying they use an "optical approximation." It does not become clear why using an approximation would make an implausible equation of motion more plausible. If I understand correctly, the authors want to limit the present paper to the optical regime and hope that in a future paper they can show the U-shape problem to be unproblematical (based on the attitude I criticized in my comment A above). This hope is way too optimistic.

RESPONSE TO REVIEWERS

Referee 1

- Comment:

I looked at the changes in the manuscript as responses to objections by another (not me) referee. In my opinion the responses are satisfying and the paper can be published.

Response:

We again thank the referee for deeming our responses to be satisfying and recommending publication.

Referee 2

- Comment:

With the present revision, I am still not enthusiastic about the paper. But my biggest concern, that prior works on relativistic Bohm trajectories were not properly credited, has been taken care of.

Response:

We thank the referee for the time given to review our manuscript. We are glad that the referee is satisfied with our crediting of our prior works.

- Comment:

The authors may benefit (for this paper or future ones) from a few more comments:

A) The authors seem to think that world lines turning around in space-time (e.g., U-shaped world lines) somehow represent particle creation (respectively, annihilation) events. But that would be a wrong treatment because a particle pair gets guided by a two-particle wave function, not by the one-particle wave function we are considering, and its probability density is a function $\rho(x_1, x_2)$ of two locations x_1 and x_2 .

Response:

The “U-shaped” worldlines are an artifact of the choice of reference frame and not due to particle creation/annihilation. The discussion of the latter case was included in the context of trajectories which emerge *when the optics approximation is not satisfied*, in which multi-particle effects become important (we did not show any of these trajectories in the paper since this regime is not consistent with our single-particle theory). This was why we suggested the need of a quantum field theory extension to multiple particles in future investigations (Page 8, Paragraph 2).

Regarding the former case, we have emphasised that a global coordinate transformation always exists which ensures the velocity field is forward-directing everywhere. The existence of such trajectories is consistent with our weak measurement-based formalism, for which it is well known that “anomalous weak values”, such as those yielding negative energies, exist.

We have added a sentence in the Discussion to make this explicit connection for readers.

- Comment:

B) The authors declare that they take the trajectories seriously as the world lines of real particles. Good. But in view of the emphasis on operational considerations, half of all readers will think that the trajectories are just a mathematical fiction and not real.

Response:

The emphasis on operational considerations was intended to convince readers that the plotted trajectories are consistent with an interpretation of these worldlines as belonging to real particles. This is one of the main advances of our paper in contrast to some of the prior works cited by the referees in previous discussions (e.g. by Bohm, Nikolic etc.) which make no connection to such a measurement model.

- Comment:

C) The authors dismiss the U-shape problem saying they use an “optical approximation.” It does not become clear why using an approximation would make an implausible equation of motion more plausible. If I understand correctly, the authors want to limit the present paper to the optical regime and hope that in a future paper they can show the U-shape problem to be unproblematical (based on the attitude I criticized in my comment A above). This hope is way too optimistic.

Response:

The optical approximation is a matter of internal consistency within our paper which focuses on describing single particle dynamics. By construction this neglects multi-particle/particle creation and annihilation effects, which would need a corresponding multi-particle theory.

We again thank the referees and Dr. Bentivegna for taking the time to review our manuscript. We list below all the revisions made to our manuscript (also highlighted blue in the final revision).

- Removed terminology of “new” or “novel” as per *Nature Communications* policy (see Pages 1-3).
- Added a sentence in the Discussion connecting regions of negative density due to boosts with anomalous weak values.
- Rectified minor typos in the proof on Pages 11-12.
- Made minor aesthetic changes to Figs. 2-6.

With kind regards,
Joshua Foo
(On behalf of all authors)

joshua.foo@uq.edu.au

+61 432 536 832

ARC Centre for Quantum Computation and Communication Technology (CQC2T)
University of Queensland,
Australia